# The Effect of Instrumental Stray Light on Brewer and Dobson Total Ozone Measurements

Omid Moeini[1,2], Zahra Vaziri Zanjani[1], C. Thomas McElroy[1], David W. Tarasick[2], Robert D. Evans[3], Irina Petropavlovskikh[3], and Keh-Harng Feng[1]

[1]Department of Earth and Space Science and Engineering, Lassonde School of Engineering, and the Centre for Research in Earth and Space Science, York University, Toronto, ON, Canada.
[2]Air Quality Research Division, Environment and Climate Change Canada, Toronto, ON, Canada.
[3]Global Monitoring Division, NOAA Earth System Research Laboratory, Boulder, CO, USA.

*Correspondence to*: Omid Moeini (omidmns@yorku.ca)

**Abstract.** Dobson and Brewer spectrophotometers are the primary, standard instruments for ground-based ozone measurements under the World Meteorological Organization's (WMO) Global Atmosphere Watch program. The accuracy of the data retrieval for both instruments depends on a knowledge of the ozone absorption coefficients and some assumptions underlying the data analysis. Instrumental stray light causes non-linearity in the response of both the Brewer and Dobson to ozone at large ozone slant paths. In addition, it affects the effective ozone absorption coefficients and extraterrestrial constants that are both instrument dependent. This effect has not been taken into account in the calculation of ozone absorption coefficients that are currently recommended by WMO for the Dobson network. The ozone absorption coefficients are calculated for each Brewer instrument individually, but in the current procedure the effect of stray light is not being considered. This study documents the error caused by the effect of stray light in the Brewer and Dobson total ozone measurements using a physical model for each instrument. For the first time, new ozone absorption coefficients are calculated for the Brewer and Dobson instruments taking into account the stray light effect. The analyses show that the differences detected between the total ozone amounts deduced from Dobson AD and CD pair wavelengths are related to the level of stray light within the instrument. The discrepancy introduced by the assumption of a fixed height for the ozone layer for ozone measurements at high latitude sites is also evaluated. The ozone data collected by two Dobson instruments during the period of December 2008 to December 2014 are compared with ozone data from a collocated double monochromator Brewer spectrophotometer (Mark III). The results illustrate the dependence of Dobson AD and CD pair measurements on stray light.

## 1 Introduction

Routine atmospheric total column ozone measurements started in the mid-1920s with a Féry spectrophotometer (Dobson, 1931). Following the International Geophysical Year (1958) a worldwide network was developed with a number of Dobson instruments that were installed around the world to monitor total ozone variations. In the early 1980s the automated Brewer became commercially available (Kerr et al., 1981). A network was also introduced for the Brewer as observing organizations

started to use these instruments alongside the Dobson for long-term measurements. Although the principle behind the measurements of the Brewer and Dobson instruments is generally the same, seasonal and systematic differences in respective TOC (Total Ozone Column) products became evident after long-term co-incident measurements were accumulated (Staehelin et al., 1998; Vanicek, 2006). The adoption of the Bass and Paur (1985) ozone cross-sections (BP) for the Dobson instrument

in 1992 put both instruments on the same reference scales (the Brewer uses BP) and reduced the difference to 4 % (Kerr et al., 1988) but it did not resolve the seasonal and –non-seasonal differences (Vanicek, 2006).

Temperature corrections to the ozone absorption cross-sections may reduce the systematic errors of Dobson ozone data by up to 4 % (Bernhard et al., 2005). The seasonal differences between the measurements by the two instrument types are related to the ozone effective temperature, which affects differently the ozone absorption measured by the Brewer and Dobson

instruments (Bernhard et al., 2005; Kerr et al., 1988; Scarnato et al., 2009; Van Roozendael et al., 1998; Vanicek, 2006) because of the different wavelengths employed for the measurements. The impact of different laboratory-determined ozone cross-sections has also been investigated and showed up to a 3 % change for the Brewer and 1 % for Dobson data (Redondas et al., 2014).

To facilitate the replacement of Dobson instruments with Brewers, statistical methods have been developed to derive transfer

functions for converting Dobson measurements to the Brewer scale (Staehelin et al., 2003). These methods have been partly successful, but they cannot entirely explain the differences between the measurements of the two instruments (Scarnato et al., 2010). Scarnato et al. (2010) found an unexplained 3 % drift over a 10-year period (1988-1997) between Arosa's Dobson and Brewer total ozone series.

Analysis of the data obtained by the Dobson at the South Pole showed that the assumption of the ozone layer being at a fixed

height leads to an error in the air mass calculation. The errors caused by this assumption may exceed 4 % in ozone measurements when the ozone distribution is distorted by the "ozone hole" (Bernhard et al., 2005).

The stray light has been demonstrated to affect measurements by both instrument types (Bais et al., 1996). Basher (1982) used a mathematical model to estimate stray light levels present in the measurements of a particular Dobson instrument. According to Basher (1982), errors of 1, 3 and 10% may be present at air mass values of 2.5, 3.2 and 3.8, respectively for direct sun AD

measurements. Christodoulakis et al. (2015) employed Basher's model to estimate the stray light level of Dobson #118 at the Athens Dobson station using direct sun AD wavelength pair measurements collected over a large range of solar zenith angles. The result showed that the mean underestimation of ozone was 3.5 DU (or about 1% of the station's mean total ozone column value) for measurements with air mass values of up to 2.5. However, a single-pair parameter was not found for Basher's model that succeeded in calculating the stray light correction for all experimental days. Christodoulakis et al. (2015) concluded that

Basher's model cannot quantify the effect of stray light on TOC measurements made by the Dobson instrument under all conditions and that further study was needed. This was also noted by Basher (1982) and Evans et al. (2009).

Karppinen et al. (2014) employed the method suggested by Kiedron et al. (2008) to correct the data collected by a single monochromator Brewer during an Intercomparison/Calibration campaign for Nordic Brewers and Dobsons held at Sodankylä 8–24 March 2011 and a follow up campaign at Izaña observatory, Tenerife, between 28 October and 18 November 2011. The

method suggested relies heavily on the dispersion information for the instrument which is not available for all instruments, especially in the historical record.

The errors caused by stray light are particularly significant at high latitudes in the late winter and early spring when measurements are made at large SZA and large TOC. Such errors are of considerable importance if those data are to be used

5 for trend analysis or satellite data validation. In particular, if such data are used in cases where Dobsons or single monochromator Brewers are replaced by instruments with a significantly lower level of stray light, such as double monochromator Brewers (Mark III), a significant false positive trend in ozone may result.

The main goal of this study is to investigate and document these sources of error in total ozone as measured by the Dobson and Brewer instruments at high latitudes.

10 **2 Method**

**2.1 Retrieval Algorithm**

According to the Beer-Lambert law, the spectral irradiance $I(\lambda)$ from a direct solar spectrum at the Earth's surface can be expressed as:

15 $I(\lambda) = I_0(\lambda)exp(-\tau(\lambda))$            (1)

where $\tau(\lambda)$ is the optical thickness of the incident path and $I_0(\lambda)$ is the extraterrestrial irradiance at wavelength $\lambda$

$\tau(\lambda) = \alpha(\lambda)X\mu + \beta(\lambda)\frac{P_s}{P_0}m_R + \delta(\lambda)m_a$       (2)

20 And

$\alpha(\lambda)$   - Monochromatic ozone absorption coefficient at wavelength $\lambda$

$X$    - Total ozone column (TOC)

$\mu$    - Relative optical air mass corresponding to ozone absorption

$\beta(\lambda)$   - Rayleigh optical depth for a one-atmosphere path

25 $P_s$    - Station pressure

$P_0$    - Mean sea level pressure (101.325 kPa)

$m_R$   - Relative optical air mass corresponding to Rayleigh scattering (extinction)

$\delta(\lambda)$   - Aerosol optical depth

$m_a$   - Relative optical air mass corresponding to aerosol scattering (extinction)

The Dobson spectrophotometer does not measure the intensity of sunlight at a single wavelength but instead determines the ratio between the irradiance at two wavelengths, one strongly absorbed and the other more weakly affected by ozone. Several wavelength pairs are used by the Dobson algorithm for calculating total column ozone. In order to minimize the effect of aerosol and other absorbers, two wavelength pairs are used such as AD, AC or CD where the A pair is (305.5 / 325.4 nm), C is (311.5 / 332.4 nm) and D is (317.6 / 339.8 nm) (Evans and Komhyr, 2008; Komhyr et al., 1993). For example, the total ozone using AD wavelength pair observations is retrieved by following expression:

$$X = (N_A - N_D - K - \Delta\beta_{AD}\frac{P_s}{P_0}m_R - \Delta\delta_{AD}m_a)/(\mu\Delta\bar{\alpha}_{AD}) \tag{3}$$

$$N_A = ln[I_0(305.5)/I_0(325.4)] - ln[I(305.5)/I(325.4)] \tag{4}$$

$$N_D = ln[I_0(317.6)/I_0(339.8)] - ln[I(317.6)/I(339.8)] \tag{5}$$

$$\Delta\beta_{AD} = [\beta(305.5) - \beta(325.4)] - [\beta(317.6) - \beta(339.8)] \tag{6}$$

$$\Delta\delta_{AD} = [\delta(305.5) - \delta(325.4)] - [\delta(317.6) - \delta(339.8)] \tag{7}$$

$$\Delta\bar{\alpha}_{AD} = [\bar{\alpha}(305.5) - \bar{\alpha}(325.4)] - [\bar{\alpha}(317.6) - \bar{\alpha}(339.8)] \tag{8}$$

where $\Delta\bar{\alpha}$ is the effective differential ozone absorption coefficient at -46.3º C and $K$ is the instrument constant. Other double wavelength pairs such as CD can be used for the ozone calculation by modifying Eq. (3) accordingly (Komhyr et al., 1993). The basic measurement principle for the Brewer instrument is the same as the Dobson. However, the Brewer measures the intensity of four operational wavelengths quasi-simultaneously. The total ozone is calculated using the following equation:

$$X = (MS9 + \Delta\beta\frac{P_s}{P_0}m_R - ETC)/(\Delta\bar{\alpha}\mu) \tag{9}$$

where $\Delta\bar{\alpha}$ and $ETC$ are the effective differential ozone absorption coefficient at -45º C and Extra-Terrestrial Constant (ETC) respectively. Both are obtained from a linear weighted combination of the logarithms of their individual values at the four wavelengths used for the total ozone retrieval (Kerr, 2002). $MS9$ is calculated from a linear combination of the logarithms of the intensities $(I(\lambda_i))$ measured at the four wavelengths $\lambda_i = (310.0, 313.5, 316.8, 320.0)$, multiplied by weighting coefficients $w_i$.

$$MS9 = \sum_{i=1}^{4} w_i . lnI(\lambda_i) = ln[I(310.0)] - 0.5\, ln[I(313.5)]$$

$$-2.2\, ln[I(316.8)] + 1.7\, ln[I(320.0)] \tag{10}$$

$$\Delta\bar{\alpha} = \sum_{i=1}^{4} w_i . \bar{\alpha}(\lambda_i) \tag{11}$$

$$\Delta\beta = \sum_{i=1}^{4} w_i . \beta(\lambda_i) \tag{12}$$

The weighting coefficients, $w_i$ = (1.0, -0.5, -2.2 and 1.7), have been selected to minimize the absorption of $SO_2$ and suppress any variations that change linearly with wavelength. Hence, the aerosol scattering effect, which is approximately linear with

wavelength over a narrow wavelength range, is suppressed in the calculation. The $w_i$ sum to zero, the requirement for the absorption function to be independent of absolute intensity. The ETC of a primary standard instrument is determined using observations made at Mauna Loa observatory and are calculated using the zero air mass factor extrapolation (Langley plot method). It can be transferred to other instruments by comparisons with a traveling standard instrument (Fioletov et al., 2005).

## 2.2 Effective Ozone Absorption Coefficients

To calculate the effective ozone absorption coefficients, the laboratory-determined ozone cross-sections at an effective atmospheric ozone layer temperature must be convolved with the instrument slit function, weighted by the solar flux. The BP ozone cross-sections were recommended by the International Ozone Commission ($IO_3C$) in 1992 (http://www.esrl.noaa.gov/gmd/ozwv/dobson/papers/coeffs.html) for the Brewer and Dobson networks. The calculation of the absorption coefficients, which are currently recommended by WMO for Dobson instruments, is described by Komhyr et al.

(1993) (K93 hereinafter) and the re-evaluation is described by Bernhard et al. (2005) (B05 hereinafter). Recently the $IO_3C$ has recommended the ozone cross-sections measured by Serdyuchenko et al. (2014) (SC hereinafter), as they reduce the Dobson temperature sensitivity. In this study for consistency with previous work and to be relevant and comparable to modern work both the BP and SC cross-sections are used. A correction factor

$$f_c = 1.0112 - 0.6903/[87.3 - (T - T_0)] \tag{13}$$

based on the results of Barnes and Mauersberger (1987), as suggested by K93, is used to adjust the BP cross-sections. $T$ is the temperature in kelvin and $T_0$ is 273.15 K. For this study all BP cross-sections are multiplied by this factor and calculated at -46.3 for both instruments to be consistent. This correction has been implemented in the Dobson network. For wavelengths

longer than 340 nm, where BP data are not available, the Brion et al. (1993), Daumont et al. (1992) and Malicet et al. (1995) (BDM) data are used. These data sets are available at individual temperatures and also with the associated quadratic coefficients

of temperature dependence on the IGACO (Integrated Global Atmospheric Chemistry Observations) web page. For this study the quadratic coefficients in the file 'Bp.par' are used for BP cross-sections and the Liu et al. (2007) quadratic approximation which excludes -273° K data from the quadratic temperature dependence fitting is used for BDM cross-sections. For SC cross-sections the quadratic temperature dependence fitting is also used. As the SC cross-sections are available from 300 nm, for shorter wavelengths the BP cross-sections are used. The temperature dependence of the cross-sections is expressed as:

$$\sigma(\lambda, T) = C_0(\lambda) + C_1(\lambda)T + C_2(\lambda)T^2 \tag{14}$$

where $\sigma(\lambda, T)$ is the ozone absorption cross-sections at wavelength $\lambda$ and temperature $T$, and $C_0$, $C_1$ and $C_2$ are the quadratic coefficients at wavelength $\lambda$. The absorption coefficients are calculated from the ozone cross-sections $\sigma(\lambda, T)$ and the ozone number density $\rho(z)$:

$$\alpha(\lambda) = \frac{1}{X}\int_{z_0}^{\infty} \sigma(\lambda, T(z))\rho(z)dz \tag{15}$$

where $z_0$ is the altitude of the station and $T$ is the temperature in kelvins. The total ozone column, $X$, (in Dobson unit equal to $2.69 \times 10^{16}$ ozone molecules per square centimetre) is defined as:

$$X = \frac{kT_0}{P_0}\int_{z_0}^{\infty}\rho(z)dz \tag{16}$$

where $T_0$ is 273.15 K and $k$ is the Boltzman constant (It should be noted that $X$ is used in the equations whereas TOC is used in the text). In order to account for the finite bandwidth of the Brewer and Dobson slit functions, the effective ozone absorption coefficient $\bar{\alpha}(\lambda)$ is used instead of $\alpha(\lambda)$ in the Brewer and Dobson retrieval algorithms (Basher, 1982; Vanier and Wardle, 1969):

$$\bar{\alpha}(\lambda_i) = \frac{-1}{X\mu}ln\left(\frac{\int I_0(\lambda)S(\lambda,\lambda_i)exp(-\alpha(\lambda)X\mu - \beta(\lambda)\frac{P_S}{P_0}m_R)d\lambda}{\int I_0(\lambda)S(\lambda,\lambda_i)exp(-\beta(\lambda)\frac{P_S}{P_0}m_R)d\lambda}\right) \tag{17}$$

where $S(\lambda, \lambda_i)$ is the slit function for a nominal wavelength $\lambda_i$.

The Brewer operational method employs a simpler approximation, which is identical to the approximation method of B05 and the simplest approach of K93, and also used by Redondas et al. (2014), Van Roozendael et al. (1998), Scarnato et al. (2009) and Fragkos et al. (2013):

$$\overline{\alpha}^{apx}(\lambda_i) = \frac{\int \alpha(\lambda)S(\lambda,\lambda_i)\,d\lambda}{\int S(\lambda,\lambda_i)\,d\lambda} \tag{18}$$

## 2.3 Ozone Air Mass Calculations

Both Brewer and Dobson retrievals assume a fixed height for a thin layer of ozone to calculate the ozone air mass. The following expression is used by both instruments to calculate relative optical air mass at a solar zenith angle of $\theta$:

$$\mu(\theta) = (Re + h)/[(Re + h)^2 - (Re + r)^2 sin^2\theta]^{0.5} \tag{19}$$

where $Re$ is the radius of the Earth, $r$ is the altitude of the station and $h$ is the height of the ozone layer. Using the mean Earth radius for $Re$ instead of the actual Earth radius at the station does not introduce a significant error in $\mu$. However, it is important

that the correct values for the station altitude and the height of the ozone layer are used in Eq. (19). The Dobson community has adopted a variable ozone layer height with latitude which, to some extent, is in agreement with ozone climatology, while a fixed height of 22 km is used in the Brewer network.

## 2.4 Slit Function and Stray Light Effect

Stray light is unwanted radiation from other wavelengths that arrives at the detector during measurements at a selected

wavelength. Scattering by instrument optical elements and inefficient out-of-band (OOB) rejection of the light by dispersive elements, e.g. the grating, are the main sources of stray light in the spectrometers. Particulate scattering within the instrument and radiation scattered from the atmosphere within field of view of the instrument can also contribute a stray light effect (Josefsson, 1992). Generally, holographic gratings with higher line densities generate lower stray light. The Mark II and IV versions of the Brewer demonstrate higher levels of stray light compared to the Mark III as the Mark III instruments utilize a

double monochromator with higher line density gratings that leads to significantly better rejection of the OOB light.

Since the gradient of ozone absorption is large in the ultraviolet spectral region, the stray light contribution from longer wavelengths can make up a significant fraction of the signal measured at shorter wavelengths where the intensity is reduced by ozone absorption. As the light path (air mass and ozone path) increases, stray light effects in the measurements also increase. Stray light results in an underestimated ozone column at larger ozone slant column amounts.

To characterize the stray light in an instrument it is necessary to measure the instrument slit function. The Brewer Mark III and IV can measure the wavelength range of 286.5 to 363 nm with 0.5 nm resolution. The Brewer slit function is characterized using a narrow band line source such as a laser as input source and scanning through all wavelengths. Measurements at 350 nm (not reported) have shown the slit function to be similar at all wavelengths in the Brewer measurement range. The slit function is reversed in wavelength space to account for the reciprocal nature of scanning the instrument versus scanning the

wavelength of the line source. A He-Cd laser is used commonly to measure the slit function of the Brewer (Karppinen et al., 2014; Kiedron et al., 2008; Pulli et al., 2018). For this study as well a He-Cd laser (single line at 325.029 nm) was used to

measure the slit functions of Brewers #009 and #119. Figure 1 shows the measured slit functions of Brewer Mark IV #009 (single monochromator) and Mark III #119 (double monochromator) located at Mauna Loa Observatory (MLO). Several Brewer Mark IV, Mark II, and Mark III slit functions have been measured during intercomparison campaigns (e.g. The Fourth North American Interagency Intercomparisons of Ultraviolet Monitoring near Boulder, Colorado, in 1997 and published by

Lantz et al. (2002) and Intercomparison Campaigns of the Regional Brewer Calibration Centre-Europe (RBCC-E)). The slit function consists of a core (band-pass), the shoulders, and the extended wings (Fig. 1). The stray light measured from nearby wavelengths (the wings of the slit function) is typically below $10^{-6}$ times that of the primary wavelength in the Mark III double Brewers as compared to $10^{-4}$ in the Mark II and Mark IV single Brewers. To reduce the effect of stray light, the Brewer Mark II uses a cutoff filter which strongly attenuates wavelengths longer than 345 nm. A solar blind filter (SBF) made of nickel

sulphate hexahydrate ($NiSO_4.6H_2O$) crystal sandwiched between two UV coloured glass filters (similar to Schott UG5 or UG11) is also used in the Mark IV. Figure 2 shows the transmission of a typical UG11-$NiSO_4$ filter measured by a Cary 5E spectrophotometer. The stray light level depends on the optical and mechanical configuration which is unique for each instrument, and thus two identically configured instruments can have somewhat different OOB light rejection.

In the Brewer operational wavelength calibration, the individual slit functions are characterized separately through dispersion

analysis (Gröbner et al., 1998). For this study, a symmetrical trapezoid is fitted to the measured slit functions of Brewer #009 (Single – Mark IV) and Brewer #119 (Double – Mark III) (Fig. 1). The model slit function fit to the data (red line) includes three parts: a trapezoid with nominal FWHM (Table 1) at the core band-pass, the shoulders which are modelled by fitting a Lorentzian function (Table 1) to the measured data and two horizontal straight lines for the outer parts (wings). To investigate the effect of stray light, an ideal slit function which is a trapezoid shape with a flat top at 0.87 of the full height and two straight

lines to zeros with nominal FWHM  has been used (Fig. 1, top left).

The slit functions of the world standard Dobson #83 were experimentally measured by Komhyr et al. (1993) and recently verified by Köhler et al. (2018) using a tuneable light source. However, the published slit functions are restricted to the core band-passes. It has been assumed that the Dobson instrument restricts OOB light from entering the slit. The extended wings have not been measured for the Dobson instrument. Basher (1982) attempted to estimate the level of stray light within the

Dobson instrument by fitting a mathematical model to the AD pair direct sun measurements and analysing the total column ozone changes with Solar Zenith angle. His analysis suggested that for most Dobson instruments the level of stray light is $10^{-4}$ based on the non-linearity of the AD direct sun measurements beyond an air mass factor of 3.

Another approach has been used by Evans et al. (2009) to measure the stray light entering the Dobson instrument. They used a filter that is opaque to the C-pair nominal short wavelength band-pass, and transparent outside of this range (Fig. 6 in Evans

et al. (2009)). The idea is that the filter would remove the desired band-pass from the signal and any current remaining is from OOB light. This method was used to estimate the contribution of stray light in zenith sky measurements of Dobson #65 in Boulder, CO. They also used a model approach for the stray light contribution in zenith sky measurements and concluded that the level of stray light in Dobson #65 is likely $2\times10^{-5}$.

The Dobson slit functions for short and long wavelengths are approximately a triangle with FWHM of 1.06 nm and a trapezoid with FWHM of 3.71 nm respectively (Fig. 3). For this study, symmetrical trapezoids centered at the nominal Dobson wavelengths were fitted to the experimentally determined slit functions of Dobson #83 and used as ideal slit functions. The characteristics of these trapezoids are given in Table 1. In order to account for stray light, for this study two horizontal straight lines were added to the outer parts of the ideal slit functions.

## 3 Discussion

### 3.1 Effect of stray light on ozone absorption coefficients

To calculate the ozone absorption coefficients, the standard values defined in Tables 1 and 2 are employed. To validate the calculations, the $\Delta \bar{\alpha}^{apx}$ is calculated for the double Brewer using an ideal trapezoid slit function and BP cross-sections at -45 ℃ without the Barnes and Mauersberger (1987) correction. Redondas et al. (2014) have calculated the ozone absorption coefficients for the nominal Brewer which is identical in terms of slit functions, nominal wavelengths and slit FWHMs with the double Brewer of this work using ideal trapezoid slit functions. The IGQ4 cross-sections used in Redondas et al. (2014) are the same as the BP cross-sections employed at this work. The value 0.3367 calculated using IGQ4 cross-sections at -45 ℃ (Redondas et al. Table 6) has a difference of 0.06 % with the value 0.3365 calculated here with the same cross-sections at the same temperature (-45 ℃) (Table 3).

### 3.1.1 Brewer

To be consistent with Dobson calculations, the Barnes and Mauersberger (1987) correction is applied on BP cross-sections. Both BP and SC cross-sections at -46.3 ℃ are used for calculation of $\bar{\alpha}(\lambda_i)$ and $\bar{\alpha}^{apx}(\lambda_i)$ presented in Tables 4 and 5 for the single and double Brewers.

The contribution of stray light in determining the ozone absorption coefficients can be seen from comparing the $\Delta \bar{\alpha}$ calculated using ideal slit functions (without stray light) with the values ($\Delta \bar{\alpha}$) calculated using modeled slit functions (including stray light). For the single Brewer the results show a difference of 0.7 % (modeled slit functions including stray light are less than that of the ideal slit functions) using both BP and SC cross-sections, while for the double Brewer the difference is less than 0.01 %.

### 3.1.2 Dobson

The values of $\bar{\alpha}(\lambda_i)$ and $\bar{\alpha}^{apx}(\lambda_i)$ calculated for the Dobson instrument using the modeled slit functions are provided in Table 6. $\bar{\alpha}^{adj}(\lambda_i)$ is the adjusted set of coefficients which are recommended by WMO to be used for the Dobson network. After applying the K93 data set to the observations made by World Standard Dobson Instrument #083 at Mauna Loa observatory, 0.8 % for AD pair and 2.2 % for CD pair differences in the calculated total ozone values were detected. K93 realized that

increasing $\Delta\bar{\alpha}_D$ by 2 % would decrease the discrepancies to below 0.5 %. Thus, the adjusted values were recommended by WMO to be used for the Dobson instruments.

The values of $\bar{\alpha}^{apx}(\lambda_i)$ calculated using ideal slit functions and BP cross-sections agree with the corresponding values of K93 to within ±2.0 %. In the case of $\bar{\alpha}(\lambda_i)$, the comparison indicates agreement to within ±3.4 % except for $\bar{\alpha}(339.9)$ where the
difference is about 67 %. Approximately the same difference was reported by B05 for the same wavelength compared to K93. B05 have investigated this discrepancy by using Molina and Molina (1986) cross-sections to extend the BP datasets for Dobson calculations and concluded that the K93 value for $\bar{\alpha}(339.9)$ is unreasonably high. As in B05, the calculated value, $\Delta\bar{\alpha}_D$ , presented in this study agrees better with the empirically adjusted value, $\Delta\bar{\alpha}_D^{adj}$. The comparison shows agreement to within ±1.6 % between the values calculated for this work and the K93 adjusted values.

Generally, the differences between values presented here and those from K93 are slightly higher than the difference between B05 and K93. However, it should be noted that the slit functions and the parameters used in the calculations presented here are slightly different from those used by K93 and B05 (Tables 1 and 2).
.

Clearly, the stray light level within each instrument has an effect on the ozone absorption coefficient calculations. This effect is negligible for instruments with a stray light level on the order of $10^{-5}$. But the difference could be up to 4.0 %  for AD pair coefficients ($\bar{\alpha}(\lambda_i)$ ) and 6.9 % for  CD pair coefficients  using BP cross-sections, and up to 4.6 % and 6.4 % using SC cross-sections, for instruments with levels of stray light on the order of $10^{-4}$ when compared with the values calculated using ideal slits. These differences translate to an underestimation of ozone values through Eq. (3). However, by scaling the data using
the Dobson calibration procedure, the difference between the AD measurements of the Standard instrument and a calibrated one is reduced to less than 0.7 % (Evans and Komhyr, 2008). In the Dobson AD pair calibration, scale factors are calculated for different ranges of air masses. The data from the instrument being calibrated are scaled to the data from the reference instrument. Then, using the quasi-simultaneous measurements of AD and CD pairs of the calibrated instrument a scaling factor is also calculated to reduce the CD measurements to the AD level.

**3.2 Stray light influence on low-sun measurements**

To illustrate the effect of stray light on low-sun measurements, the percentage difference between ozone derived using Brewer and Dobson retrievals with assumed constant ozone in the atmosphere are depicted as a function of ozone slant path (OSP, total ozone times air mass) in Fig. 4. As the selection of the cross-sections has insignificant effect on the results shown and discussed in this section, only BP cross-sections are plotted. Equation (1) along with parameters indicated in Table 2, are used
to model the atmosphere and calculate the solar spectrum at the surface. To retrieve the ozone values, the BP absorption coefficients calculated in Sect. 3.1 are used. To calculate the ETC values, the instrument absorption function is calculated and plotted as a function of ozone slant path, using the solar spectrum (Chance and Kurucz, 2010), Eqs. (1) and (2) and retrieval

algorithm of the Brewer (or Dobson) for an assumed constant amount ozone (325 DU in this study). The best fit to the data with air mass less than 2 (less than 3 for the Dobson instruments) is found and extrapolated to zero air mass. Figure 5 shows the best fit to data from a single Brewer. For the single Brewer the ETC is calculated as 1945.4 for a modelled trapezoid slit function with stray light, which is close to the value of 2020 calculated by Kiedron et al. (2008), noting the slight differences

in slit functions and solar spectrum. Karppinen et al. (2015) have reported 3218 for an ETC value for slit functions with stray light. However, they used LibRadtran 1.6-beta radiative transfer model to scale their modeled data to be matched with real data. For Dobsons, the ETC values calculated using ideal slits are used for other models (i.e. with $10^{-4}$ and $10^{-5}$ levels of stray light). Two versions of the Brewer are compared with the Dobson instrument measurements with two levels of stray light. AD measurements with $10^{-4}$ order of stray light show approximately 25 % discrepancy at 2000 DU OSP (air mass 6.2 in this case).

The difference is about 5 % for a typical single Brewer at the same OSP. The underestimation of total ozone as measured by the AD pair of a Dobson instrument with $10^{-5}$ level of stray light could be up to 6 % at 2000 DU OSP. It has to be noted here that AD pair measurements are conducted for air mass factors less than 2.5 and thus, during the ozone hole period (total column ozone is less than 300 DU), Dobson data will only be reported for OSP less than 750 DU (Evans and Komhyr, 2008). Evidently, the CD pair is less influenced by scattered light than the AD pair because of the smaller ozone cross-sections at the

CD wavelengths and the consequent smaller gradient with respect to wavelength in the spectrum measured. For a Dobson instrument with a minimum level of stray light ($10^{-5}$) the difference for the CD pair could be up to 1.8 % at 2000 DU OSP while it is less than 0.8 % for a typical double Brewer at the same OSP. As Dobson CD total ozone is reported for air mass values beween 2.4 and 3.5,  thus the OSP is less than 1100 DU for total ozone (TOC) less than 300 DU (Evans and Komhyr, 2008).

**3.3 Total ozone values retrieved from Dobson AD and CD pairs**

For decades the Dobson community has faced a discrepancy between the ozone values deduced from quasi-simultaneous AD and CD pair measurements. As indicated by the Dobson operational handbook, AD observations are the standard for the Dobson instrument and all other observations must, therefore, be scaled to the AD level by determining a multiplying factor. For example, the ozone values deduced from measurements on CD wavelengths should be multiplied by the factor of $X_{AD}/X_{CD}$

(where $X_{AD}$ and $X_{CD}$ are the average ozone measurements retrieved from AD and CD pairs derived from a large number of quasi-simultaneous observations covering a broad range of μ values greater than 2.0) to be scaled to those deduced from AD measurements.

Figure 6 illustrates the discrepancy in total ozone reduced from AD and CD pairs for two modeled Dobson instruments with different levels of stray light using two different ozone cross-sections (BP and SC) as a function of OSP. The ratio of the AD

to the CD pair is also shown. The adjusted coefficients calculated by K93 and recommended by WMO are used to derive the total ozone amounts for this model. The ETC values that were calculated using the Langley method for an ideal instrument are used here as well. It can be seen that, as the level of stray light increases, the difference between the AD and CD values increases, indicating the role of stray light in the observed discrepancy between AD and CD values. The results suggest that

the difference between the AD and CD values also depends on the selected cross-sections. It is apparent that the increase of the AD-CD difference as a result of increasing the level of stray light is larger for BP cross-sections than for SC cross-sections. This is consistent with Redondas et al., (2014) analysis indicating the role of ozone cross-sections in the difference detected between AD and CD measurements.

Clearly, the difference varies between Dobson instruments as it depends on the level of stray light which is unique for each individual instrument. Moreover, as discussed in Sect. 3.1, these discrepancies are reduced during calibration using simultaneous measurements with a well-maintained, standard instrument.

## 3.4 Discrepancy caused by Air Mass Calculation

To retrieve total ozone from direct sun measurements it is required that the ozone air mass value be calculated. For
measurements at the South Pole, the following values are used by the Dobson community in Eq. (19): $Re = 6356.912$, $r = 2.81$ km, and $h = 17$ km. Figure 7 shows the ratio of total ozone retrieved using air mass values calculated with four different sets of assumptions for $Re$, $r$ and $h$ to ozone values retrieved using air mass factor calculated for the South Pole employing the Dobson community's values for $Re$, $r$ and $h$ as a reference. Any discrepancy in the air mass is directly reflected in the Dobson and Brewer retrieved ozone via Eqs. (3) and (9) respectively. It is obvious that the fixed ozone layer height of 22 km, as used
by Brewer retrieval, can cause up to a 2.2 % difference at an air mass of 5.4. In addition, the altitude of the site can introduce a significant difference at high solar zenith angles.

## 4 Comparison between Dobson and Brewer measurements at the South Pole

Total ozone measurements collected by two Dobson instruments (#82 and #42,) and one double Brewer Mark III, #085, collocated and operated simultaneously at the Amundsen-Scott site (24.80º W, 89.99º S, altitude 2810 m) are used for this
comparison. Double Brewer #085 was installed at the South Pole station in 2008 and since then it was in routine operation. The Brewer data for the South Pole site are available at the WOUDC website. Due to the logistic difficulties Brewer #085 was not replaced or calibrated until 2016.

The Dobson data used for this study are freely available at: ftp://aftp.cmdl.noaa.gov/user/evans/York_Omid/. For this study all direct sun Dobson measurements are used while only one measurement representative of the day is reported to the NDACC
or WOUDC. A complete description of the South Pole dataset is provided by Evans et al., (2017). The reprocessed data using WinDobson software as described in Evans et al. (2017) are used for the analysis here. Generally, the Dobson instrument at the South Pole site is replaced with a calibrated instrument every four years. The instrument replaced is calibrated against the reference Dobson #83 and the calibration results are used to adjust and post-process the last four years of data collected at the South Pole. The calibration procedure can be found at Evans and Komhyr (2008) and the major calibration or instrument
changes regarding the South Pole dataset can be seen in Fig. 5 of Evans et al. (2017).

To avoid the impact of ozone hole on the analyses presented here only the data with total ozone values larger than 220 DU are used for this study. Quasi-simultaneous direct sun measurements performed within 5 minutes during the period of December 2008 to December 2014 were used in the present analysis. The air masses calculated by the Brewer retrieval were adjusted using Eq. (19) by applying the values used by Dobson instruments (Re = 6356.912 km, r = 2.81 km, and h = 17 km) to be consistent with the Dobson air masses. Figure 8 shows the total ozone column measured by double Brewer #085 and the total ozone retrieved using the adjusted air masses. The ratio of total ozone columns retrieved using adjusted air masses to the total ozone columns retrieved using unadjusted air masses are also shown in the same figure.

The ozone absorption coefficients calculated by Komhyr et al. (1993) and recommended by WMO have been used to retrieve ozone values for the AD and CD pairs. It is necessary to mention that only the data collected with air mass factors less than 2.5 or OSP less than 800 DU would be reported for AD pair measurements to the World Ozone and Ultraviolet Data Centre (WOUDC) or other institutes for regular research purposes. The range of air mass factor for CD measurements is 2.4 to 3.5 and that means in the case of the South Pole station, the maximum OSP would be about 1100 DU.

Figure 9 presents a comparison of the Brewer total ozone measurements with Dobson ozone observations reduced from the direct sun AD pair as a function of OSP. The ratios between the unadjusted Brewer data and Dobson measurements are also shown in the same plot. The ratio between the Brewer adjusted data and the Dobson values shows some dependence on OSP: the Dobson #82 and #42, are on average 1 % and 0.46 % higher respectively for OSPs below 800 DU. When the OSP is above 800 DU, Dobson measurements gradually become lower by up to 4 % for OSPs up to 1400 DU for Dobson #42 and up to 5 % for OSPs up to 1200 DU for Dobson number #82 (Fig. 9).

The ratio between Dobson CD pair measurements and the Brewer data also shows a dependence on OSP (Fig. 9). However, CD pair ozone values are on average 3.4 %and 4.5 %, higher for almost the entire measurement set for Dobson #82 nad #42 respectively. It should be noted that the CD pair values are scaled to the AD pair for each individual Dobson instrument. The scaling factors 1.043 and 1.025 were calculated during calibration procedure in 2015 and used for adjusting total column ozone values derived from Dobson #82 and #42 CD pair measurements respectively (private communications with R. Evans, December 2015. Note to the user: The subset of the South Pole station Dobson record used in this paper was reprocessed in June of 2015 using the WinDobson software as described in Evans et al. (2017). The NOAA archived data referenced in Evans (2017) paper (ftp://aftp.cmdl.noaa.gov/ozwv/Dobson/dobson_toSPO.txt) was finalized in 2017. It contains the same AD TOC, while CD TOC was re-processed to adjust for AD/CD differences and updates in the air mass factor calculation. The scaling factors were changed to 1.018 and 0.998 respectively and data were re-submitted to the NDACC archive (ftp://ftp.cpc.ncep.noaa.gov/ndacc/station/spole/ames/dobson/) in October 2017 - private communication with G. McConville and K. Miyagawa of NOAA, Boulder, CO, November 2018). The calibration procedures and the method for calculating the scaling factor are described and published by WMO in GAW report No. 183 (Evans and Komhyr, 2008). As illustrated by B05 (Fig. 4 of Bernhard et al. (2005)) the temperature dependence of the ozone absorption cross-sections may also cause discrepancy in Dobson total ozone measurements. As illustrated by B05 (Fig. 4 of Bernhard et al. (2005)) the temperature dependence of the ozone absorption cross-sections may also cause deviations in Dobson total ozone column measurements.

B05 assessed variability in the effective temperature using ozone profiles measured by Global Monitoring Division of NOAA (former Climate Monitoring and Diagnostics Laboratory, CMDL) at South Pole between 1991 and 2003. They found that the temperature-adjusted AD and CD ozone absorption coefficients deviated from nominal values (K93) by up to ±4 % leading to underestimation or overestimation of the total ozone column through Eq. (3). B05 also found that the 1991-2003 collection of temperature-adjusted CD ozone absorption coefficients exhibited a SZA dependence. The ozone absorption coefficient adjustment is likely magnified at low sun (large SZA) conditions (preferential for CD over AD measurements) when extreme changes in ozone and temperature profiles are observed during the ozone hole period.

It can be seen from this analysis that Dobson #42 TOC values show less dependence on OSP. Based on the physical model developed in this work, it appears that this instrument has a significantly lower level of stray light than Dobson instrument #82. As it is shown in Fig. 10, the physical model developed in this study suggests $10^{-3.5}$ and $10^{-4.1}$, level of stray light for Dobson #82 and 42, respectively.

Figure 11 shows a box plot of the difference between double-Brewer ozone measurements and Dobson values retrieved from AD and CD pairs. The data are binned for 100 DU from 400 to 2000 DU. On each box, the central red line is the median, the edges of the box are the 25th and 75th percentiles, the whiskers extend to the most extreme data points not considered outliers, and outliers are plotted individually. In each bin the values with differences larger than three standard deviations from the mean of the bin have been removed from the calculation. The number of simultaneous measurements in each bin is summarized in Table 7.

## 5 Conclusions

Physical models of the Dobson instrument and two types of Brewer spectrophotometer were developed to help better understand the effects of stray light on ozone measurements and indicate a way forward for correcting current and past data for the resulting error in the calculated ozone column. The influence of assuming a fixed ozone layer height on air mass calculations and the difference caused by this assumption to the ozone retrieval were also examined. The target accuracy for ground-based ozone measurements is 1 %, while physical models show that the stray light effect can cause a discrepancy for a typical single Brewer and Dobson AD pair at large ozone slant paths of up to 5 % and 25 %, respectively. At 2000 DU OSP the difference for a double Brewer and a Dobson CD pair with minimum level of stray light ($10^{-5}$) is up to 0.8 % and 1.8 %, respectively. This effect restricts measurements at high latitudes, such as polar stations, particularly in the late winter and early spring when the ozone slant column is particularly large. It is considerably important if those data are to be used for trend analysis. In particular, if such a data is used in the case where Dobsons or single Brewers are replaced by double Brewers, a significant false positive trend in ozone may result.

Stray light also can affect the calculation of ozone absorption coefficients. Currently, an approximation method is used to calculate the absorption coefficients for Brewer instruments. The analysis shows that using a modeled trapezoid slit function

with stray light (instead of an idealized trapezoidal one) and taking into account the solar spectrum, leads to a difference of 0.7 % and less than 0.01 % in calculated coefficients for a typical single and double Brewer, respectively.

Absorption coefficients for the Dobson spectrophotometers, taking into account the effect of stray light, have been calculated and compared with the results of similar calculations by K93, which are the coefficients recommended by WMO. The slit functions of Dobson #83 have been measured using a tunable light source (K93). Recently, the measured slit functions and calculated coefficients were verified by measuring the slit functions of three Dobsons (#74, #64, and #83) at the Physikalisch-Technische Bundesanstalt (PTB) in Braunschweig in 2015 and at the Czech Metrology Institute (CMI) in Prague in 2016 within the EMRP ENV 059 project "Traceability for atmospheric total column ozone" (Köhler et al., 2018). Köhler et al. (2018) showed that the optical properties of these three Dobsons deviate from the specification described by G. M. B. Dobson . However, the AD pair ozone absorption coefficents derived from new slit functions lead to less than 1 % deviation in total ozone column values. It is generally assumed that the slit functions of other Dobson spectrophotometers are similar to the standard one (#83).

For the study presented here, the slit functions were modeled to examine the effect of stray light on the calculation of ozone absorption coefficients. The results show that $10^{-5}$ level of stray light has negligible effect on absorption coefficient calculations while the difference could be up to 4.0 % and 4.6 % for AD coefficients and up to 6.9 % and 6.4 % for CD coefficients for an instrument with $10^{-4}$ level of stray light using BP and SC cross-sections, respectively.

When quasi-simultaneous measurements are made using Dobson AD and CD wavelengths, the results may not agree. For decades the Dobson community has faced such differences. The AD pair is the standard for Dobson measurements and the observations using other pairs' data should be scaled to the AD pair before release. Our analysis indicates that the difference between quasi-simultaneous measurements using AD and CD pairs is related to the level of stray light inside each Dobson instrument. Higher levels of stray light lead to larger differences between the values deduced from AD and CD wavelengths. However, this increase in difference between AD and CD values is slightly smaller when the SC cross-sections are used than using the BP cross-sections.

Both Brewer and Dobson retrievals assume a fixed height for the ozone layer to calculate the ozone air mass. A fixed height of 22 km is used by the Brewer network for all sites while a variable ozone layer height changing with latitude is employed by the Dobson community. The assumption of a 22 km height for the ozone layer at the South Pole, compared to the 17 km height used in the Dobson analysis, leads to a 2.2 % difference in ozone column at an air mass of 5.4.

Comparisons with total ozone data larger than 220 DU from a double Mark III Brewer spectrophotometer located at the South Pole indicate some dependence on OSP for the Dobson measurements. For the OSPs below 800 DU the AD vales are generally 1 % higher. However, for OSPs between 800 and 1400 DU the Dobson AD measurements are lower by up to 4 %. Using the physical model developed in this study the stray light levels of Dobson #82 and #42 are estimated as $10^{-3.5}$ and $10^{-4.1}$, respectively.

Observations made at the CD wavelengths also show some dependence on OSP. Compared to Brewer data, the CD values are, on average, 4 % higher for almost the entire range of measurements. However, as is the case for the AD pair, the CD pair

values also decrease at larger OSPs. It should be noted that the Dobson AD and CD pair measurements are not reported for air mass factors above 2.5 and 3.5 respectively due to the effect of stray light.

## Acknowledgements

The Authors thank the many NOAA/ESRL technical staff members and Officers for their invaluable efforts to operate the facility year-round at the South Pole and obtain the data used in this report. The U.S. National Science Foundation provides the infrastructure and logistics support for the facility at the South Pole. The Brewer data were obtained from the World Ozone and Ultraviolet Radiation Data Centre (WOUDC, http://www.woudc.org) operated by Environment and Climate Change Canada, Toronto, Ontario, Canada, under the auspices of the World Meteorological Organization. The first author is grateful for support from Environment and Climate Change Canada and the Natural Sciences and Engineering Research Council of Canada (NSERC).

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

**Table 1: Brewer and Dobson optical parameters**

|  | Brewer | Dobson |
|---|---|---|
| Nominal Wavelengths (nm) | 310.05, 313.50, 316.80, 320.00 | A: 305.5/325.0, C: 311.5/332.40, D: 317.5/339.9 |
| Slit Function | Single: trapezoid at centre 0.539, 0.555, 0.545, 0.538 nm FWHM, Lorentzian fitted to the measured slit of #009 for shoulders Double: trapezoid at centre 0.539, 0.555, 0.545, 0.538 nm FWHM, Lorentzian fitted to the measured slit of #119 for shoulders | Short Channels: trapezoid 1.66, 1.84, 2.02 nm at the base and 0.16, 0.16, 0.16 nm at the top Long Channels: trapezoid 4.60, 5.40, 5.75 nm at the base and 2.35, 2.50, 2.50 nm at the top |
| Stray light level | Single: ~ $1 \times 10^{-4}$ Double: ~ $1 \times 10^{-6}$ | $1 \times 10^{-5}$ and $1 \times 10^{-4}$ |
| Filters | Single: UG-11 and NiSO4 filters – Zero below 280 and above ~330 nm Double: PMT sensitivity - Zero below 250nm and above 800 nm | Cobalt filter - zero above ~360 nm |

**Table 2: Parameters for calculation of ozone absorption coefficients.**

| Parameter | Komhyr et al. (1993) | This Work |
|---|---|---|
| Slit function | Measured slit functions (Fig. 1 of Komhyr et al. (1993)) | Dobson: Parameterized from Fig. 1 of Komhyr et al. (1993) (Details in Table 1) Brewer: Parameterized from laser scan of #009 and #119 (Details in Table 1) |
| Effective temperature | -46.3°C | -46.3°C |
| Solar spectrum | Furukawa et al. (1967) | Chance and Kurucz (2010)[*] |
| Ozone cross-sections | Bass and Paur (1985) | Bass and Paur (1985) and Serdyuchenko et al., (2014) |
| Rayleigh Scattering | Bates (1984) | Bates (1984) |
| Ozone profile | Bhartia et al. (1985) for 45° N and 325 DU | Bhartia et al. (1985) for 45° N and 325 DU |
| Air mass | 2 | 2 |

[*] For this study the wavelength range of 285-363 nm is used for calculations.

**Table 3: Ozone absorption coefficients calculated here and the value calculated by Redondas et al. (2014)**

| | | $\bar{\alpha}^{apx}(\lambda_i)$ (atm cm$^{-1}$) calculated for Double Brewer using ideal slits and BP cross-sections at -45 ºC without Barnes (1987) correction | From Redondas (2014) Table 6; effective ozone absorption coefficient calculated using IQG4 B&P cross-sections |
|---|---|---|---|
| | | Ideal (trapezoid) | Ideal (trapezoid) |
| Wavelength (nm) | FWHM (nm) | $\bar{\alpha}^{apx}(\lambda_i)$ | $\bar{\alpha}^{apx}(\lambda_i)$ |
| 310.05 | 0.539 | 1.0044 | |
| 313.50 | 0.555 | 0.6793 | |
| 316.80 | 0.545 | 0.3760 | |
| 320.00 | 0.538 | 0.2935 | |
| $\Delta\bar{\alpha}^{apx}$ | | 0.3365 | 0.3367 |

**Table 4: Single Brewer ozone absorption coefficients**

| | | BP cross-sections | | | | SC cross-sections | | | |
|---|---|---|---|---|---|---|---|---|---|
| | | Ideal | | Model (with Stray light) | | Ideal | | Model (with Stray light) | |
| Wavelength (nm) | FWHM (nm) | $\bar{\alpha}^{apx}(\lambda_i)$ | $\bar{\alpha}(\lambda_i)$ | $\bar{\alpha}^{apx}(\lambda_i)$ | $\bar{\alpha}(\lambda_i)$ | $\bar{\alpha}^{apx}(\lambda_i)$ | $\bar{\alpha}(\lambda_i)$ | $\bar{\alpha}^{apx}(\lambda_i)$ | $\bar{\alpha}(\lambda_i)$ |
| 310.05 | 0.539 | 1.0087 | 1.0127 | 1.0141 | 1.0102 | 1.0012 | 1.0167 | 1.0068 | 1.0142 |
| 313.50 | 0.555 | 0.6824 | 0.6842 | 0.6828 | 0.6833 | 0.6752 | 0.6825 | 0.6757 | 0.6817 |
| 316.80 | 0.545 | 0.3774 | 0.3789 | 0.3768 | 0.3789 | 0.3706 | 0.3751 | 0.3700 | 0.3751 |
| 320.00 | 0.538 | 0.2944 | 0.2962 | 0.2923 | 0.2959 | 0.2973 | 0.2996 | 0.2952 | 0.2993 |
| $\Delta\bar{\alpha}^{apx} or \Delta\bar{\alpha}$ | | 0.3377 | 0.3406 | 0.3407 | 0.3380 | 0.3538 | 0.3596 | 0.3566 | 0.3570 |

15           *BP cross-sections with Barnes (1987) correction; Both BP and SC cross-sections at -46.3.

**Table 5: Double Brewer ozone absorption coefficients**

| Wavelength (nm) | FWHM (nm) | BP cross-sections | | | | SC cross-sections | | | |
|---|---|---|---|---|---|---|---|---|---|
| | | Ideal | | Model (with Stray light) | | Ideal | | Model (with Stray light) | |
| | | $\bar{\alpha}^{apx}(\lambda_i)$ | $\bar{\alpha}(\lambda_i)$ | $\bar{\alpha}^{apx}(\lambda_i)$ | $\bar{\alpha}(\lambda_i)$ | $\bar{\alpha}^{apx}(\lambda_i)$ | $\bar{\alpha}(\lambda_i)$ | $\bar{\alpha}^{apx}(\lambda_i)$ | $\bar{\alpha}(\lambda_i)$ |
| 310.05 | 0.539 | 1.0087 | 1.0127 | 1.0089 | 1.0126 | 1.0012 | 1.0167 | 1.0014 | 1.0166 |
| 313.5 | 0.555 | 0.6824 | 0.6842 | 0.6826 | 0.6841 | 0.6752 | 0.6825 | 0.6754 | 0.6825 |
| 316.8 | 0.545 | 0.3773 | 0.3789 | 0.3776 | 0.3789 | 0.3706 | 0.3751 | 0.3708 | 0.3751 |
| 320 | 0.538 | 0.2947 | 0.2962 | 0.2950 | 0.2962 | 0.2974 | 0.2996 | 0.2976 | 0.2996 |
| $\Delta\bar{\alpha}^{apx} or \Delta\bar{\alpha}$ | | 0.3384 | 0.3406 | 0.3384 | 0.3405 | 0.3538 | 0.3595 | 0.3538 | 0.3594 |

*BP cross-sections with Barnes (1987) correction; Both BP and SC cross-sections at -46.3.

**Table 6: Dobson wavelengths and Ozone Absorption coefficients**

| | Effective Ozone Absorption Coefficient (atm cm)$^{-1}$ | | | | | | | | | | | | | | |
|---|---|---|---|---|---|---|---|---|---|---|---|---|---|---|---|
| | Komhyr et al. (1993) | | | This Work | | | | | | | | | | | |
| | | | | BP cross-sections | | | | | | SC cross-sections | | | | | |
| | | | | Model (Ideal) | | Model (10$^{-5}$)* | | Model (10$^{-4}$) | | Model (Ideal) | | Model (10$^{-5}$)* | | Model (10$^{-4}$) | |
| Wavelength, nm or pair | $\bar{\alpha}^{apx}(\lambda_i)$ | $\bar{\alpha}(\lambda_i)$ | $\bar{\alpha}^{adj}(\lambda_i)$ | $\bar{\alpha}^{apx}(\lambda_i)$ | $\bar{\alpha}(\lambda_i)$ | $\bar{\alpha}^{apx}(\lambda_i)$ | $\bar{\alpha}(\lambda_i)$ | $\bar{\alpha}^{apx}(\lambda_i)$ | $\bar{\alpha}(\lambda_i)$ | $\bar{\alpha}^{apx}(\lambda_i)$ | $\bar{\alpha}(\lambda_i)$ | $\bar{\alpha}^{apx}(\lambda_i)$ | $\bar{\alpha}(\lambda_i)$ | $\bar{\alpha}^{apx}(\lambda_i)$ | $\bar{\alpha}(\lambda_i)$ |
| 305.5 | 1.917 | 1.915 | | 1.912 | 1.930 | 1.913 | 1.929 | 1.867 | 1.870 | 1.919 | 1.946 | 1.919 | 1.945 | 1.866 | 1.879 |
| 325 | 0.115 | 0.109 | | 0.114 | 0.111 | 0.114 | 0.111 | 0.119 | 0.111 | 0.113 | 0.111 | 0.114 | 0.111 | 0.118 | 0.111 |
| A | 1.802 | 1.806 | 1.806 | 1.799 | 1.819 | 1.799 | 1.818 | 1.748 | 1.759 | 1.805 | 1.835 | 1.805 | 1.833 | 1.748 | 1.768 |
| 311.5 | 0.87 | 0.873 | | 0.867 | 0.879 | 0.868 | 0.879 | 0.848 | 0.846 | 0.867 | 0.885 | 0.868 | 0.885 | 0.850 | 0.854 |
| 332.4 | 0.039 | 0.04 | | 0.041 | 0.042 | 0.041 | 0.042 | 0.042 | 0.042 | 0.041 | 0.041 | 0.041 | 0.041 | 0.042 | 0.042 |
| C | 0.831 | 0.833 | 0.833 | 0.826 | 0.838 | 0.827 | 0.838 | 0.806 | 0.804 | 0.826 | 0.843 | 0.827 | 0.843 | 0.809 | 0.812 |
| 317.5 | 0.379 | 0.384 | | 0.383 | 0.390 | 0.384 | 0.390 | 0.393 | 0.387 | 0.377 | 0.387 | 0.379 | 0.387 | 0.388 | 0.384 |
| 339.9 | 0.01 | 0.017 | | 0.010 | 0.010 | 0.010 | 0.010 | 0.010 | 0.011 | 0.010 | 0.011 | 0.010 | 0.011 | 0.011 | 0.011 |
| D | 0.369 | 0.367 | 0.374 | 0.373 | 0.380 | 0.374 | 0.380 | 0.382 | 0.376 | 0.367 | 0.377 | 0.368 | 0.376 | 0.377 | 0.373 |
| AD | 1.433 | 1.439 | 1.432 | 1.426 | 1.439 | 1.425 | 1.438 | 1.366 | 1.383 | 1.438 | 1.458 | 1.437 | 1.457 | 1.370 | 1.394 |
| CD | 0.462 | 0.466 | 0.459 | 0.453 | 0.458 | 0.452 | 0.458 | 0.424 | 0.428 | 0.459 | 0.467 | 0.459 | 0.467 | 0.432 | 0.439 |

* The numbers inside the braces are showing the levels of stray light.

**Table 7: The number of simultaneous measurements in each bin**

|  | Dobson #82 | | Dobson #42 | |
| --- | --- | --- | --- | --- |
| Bins (OSP) | AD | CD | AD | CD |
| [400 500) | 0 | 0 | 0 | 0 |
| [500 600) | 17 | 22 | 2 | 0 |
| [600 700) | 135 | 71 | 37 | 18 |
| [700 800) | 402 | 288 | 229 | 149 |
| [800 900) | 147 | 195 | 130 | 154 |
| [900 1000) | 150 | 109 | 22 | 44 |
| [1000 1100) | 89 | 123 | 52 | 24 |
| [1100 1200) | 4 | 44 | 46 | 63 |
| [1200 1300) | 0 | 39 | 36 | 40 |
| [1300 1400) | 0 | 42 | 4 | 47 |
| [1400 1500) | 0 | 36 | 0 | 19 |
| [1500 1600) | 0 | 17 | 0 | 4 |
| [1600 1700) | 0 | 6 | 0 | 1 |
| [1700 1800) | 0 | 9 | 0 | 1 |
| [1800 1900) | 0 | 0 | 0 | 10 |
| [1900 2000) | 0 | 0 | 0 | 0 |
| Total | 944 | 1001 | 558 | 574 |

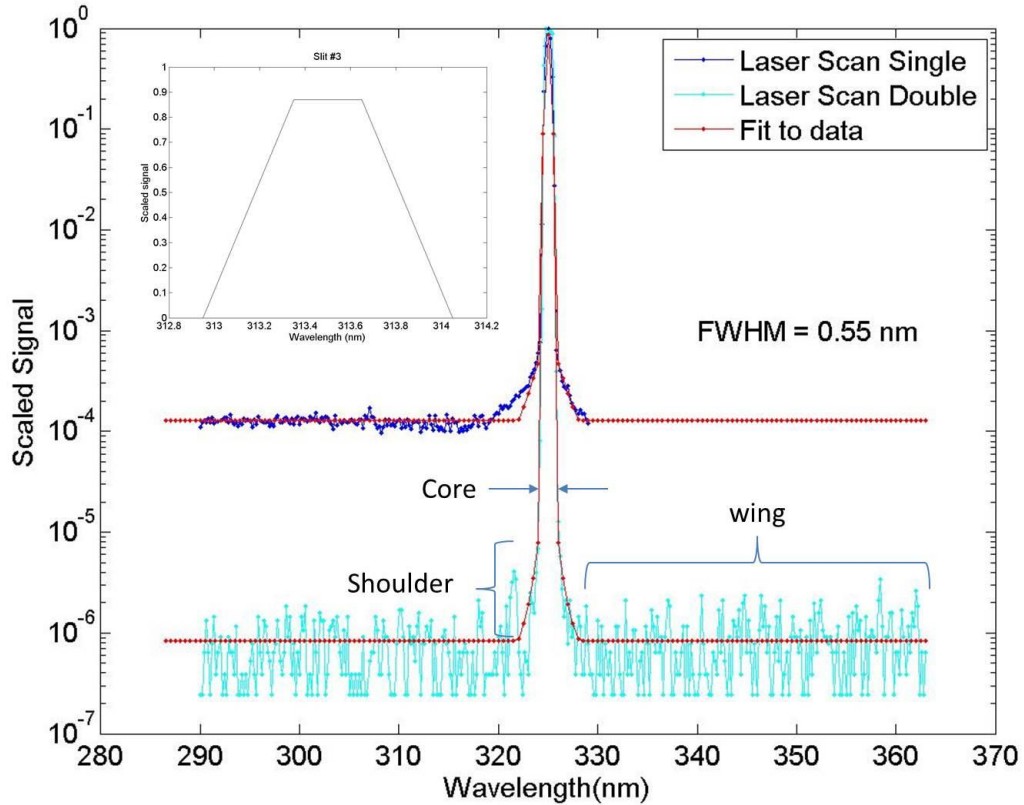

**Figure 1: Slit functions measured with a He-Cd laser for single #009 and double #119 Brewers at Mauna Loa as well as fitted models; The ideal slit function is also shown inside the main graph.**

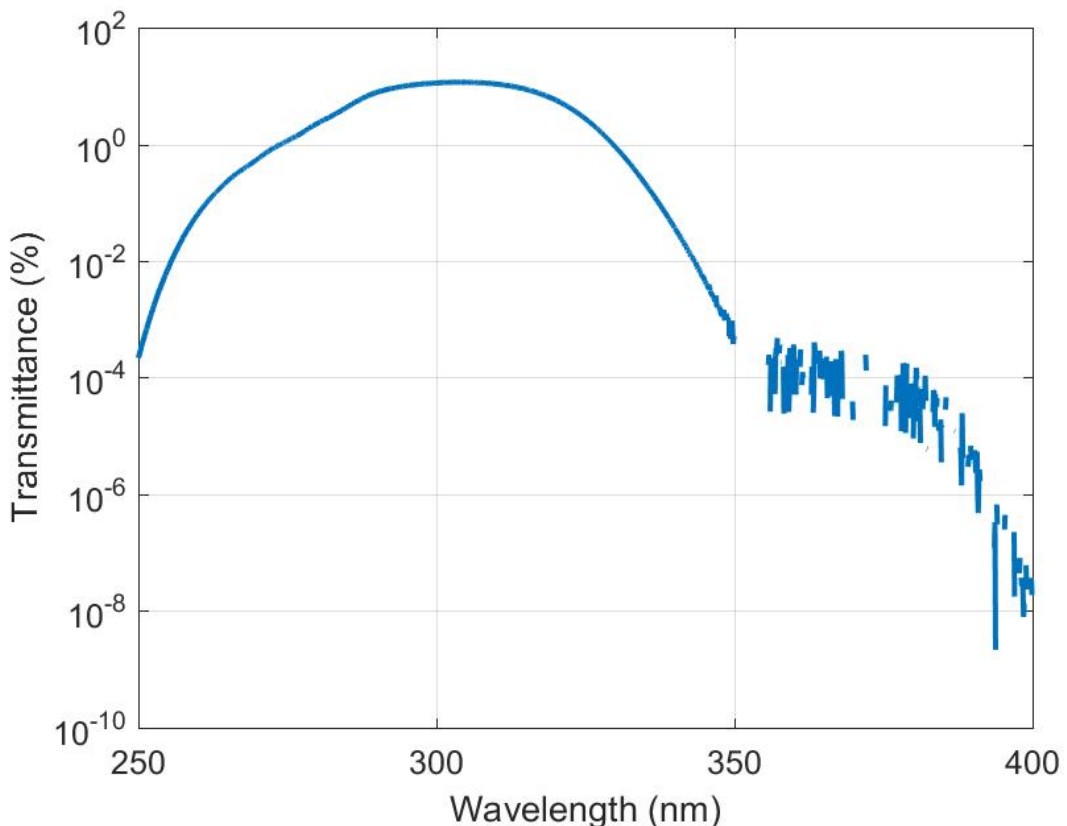

**Figure 2: The transmission of a typical combined UG11-NiSO₄ filter utilized by Brewer Mark IV to reduce the stray light measured with a Cary 5E spectrophotometer for Brewer #154 filters.**

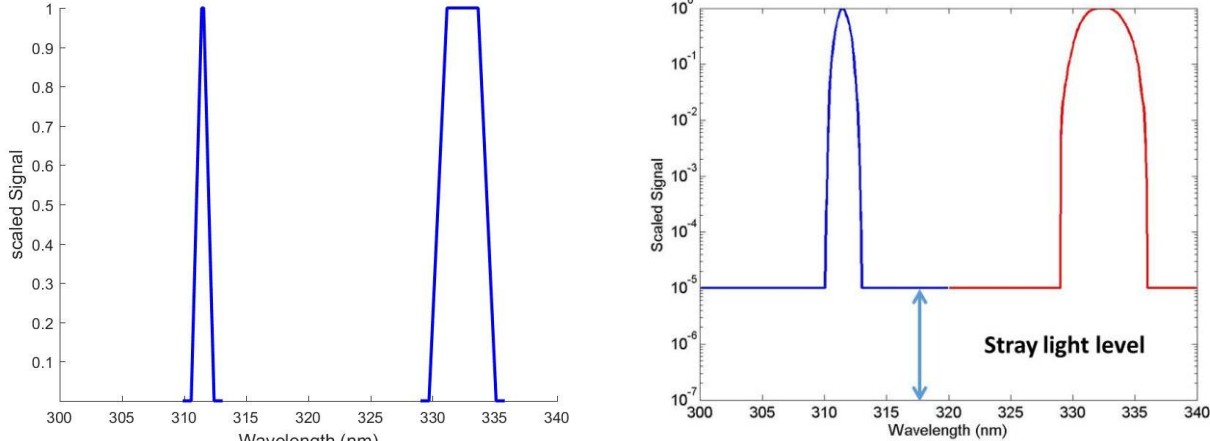

**Figure 3: Dobson C-pair ideal slit functions (Left) parameterised from Figure 1 of Komhyr 1993. Dobson Modelled slit functions (right). Two straight lines have been added to the core slit functions in order to account for stray light.**

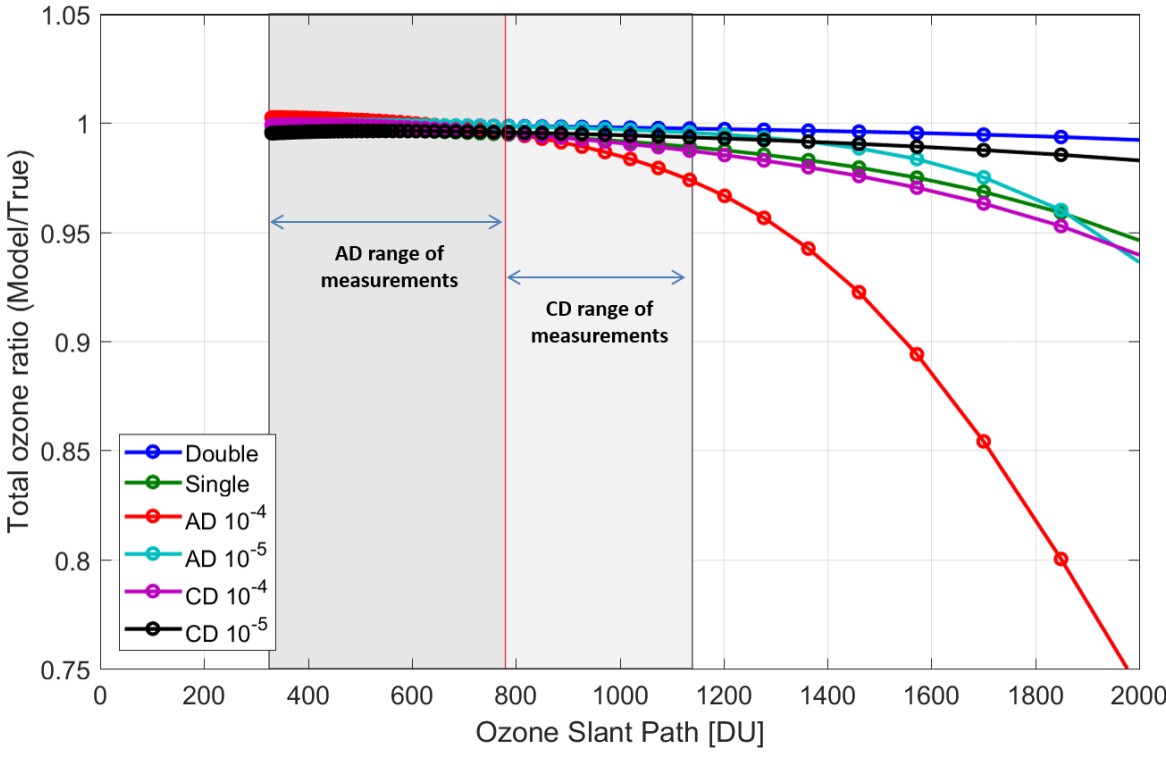

**Figure 4: Ratio of the values retrieved from modeled single and double Brewers as well as a Dobson instrument with different levels**
10  **of stray light (the numbers in front of the pairs are showing the levels of stray light) to assumed 325 DU  ozone in the atmosphere (See text for details). The calculations reported in this study for the BP absorption coefficients have been used to retrieve total ozone. It should be noted that the air mass factor range recommended for AD measurements is 1.015 to 2.5 or less than 800 DU OSP and for CD measurements is 2.4 to 3.5 or less than 1200 DU OSP** (Evans and Komhyr, 2008)**.**

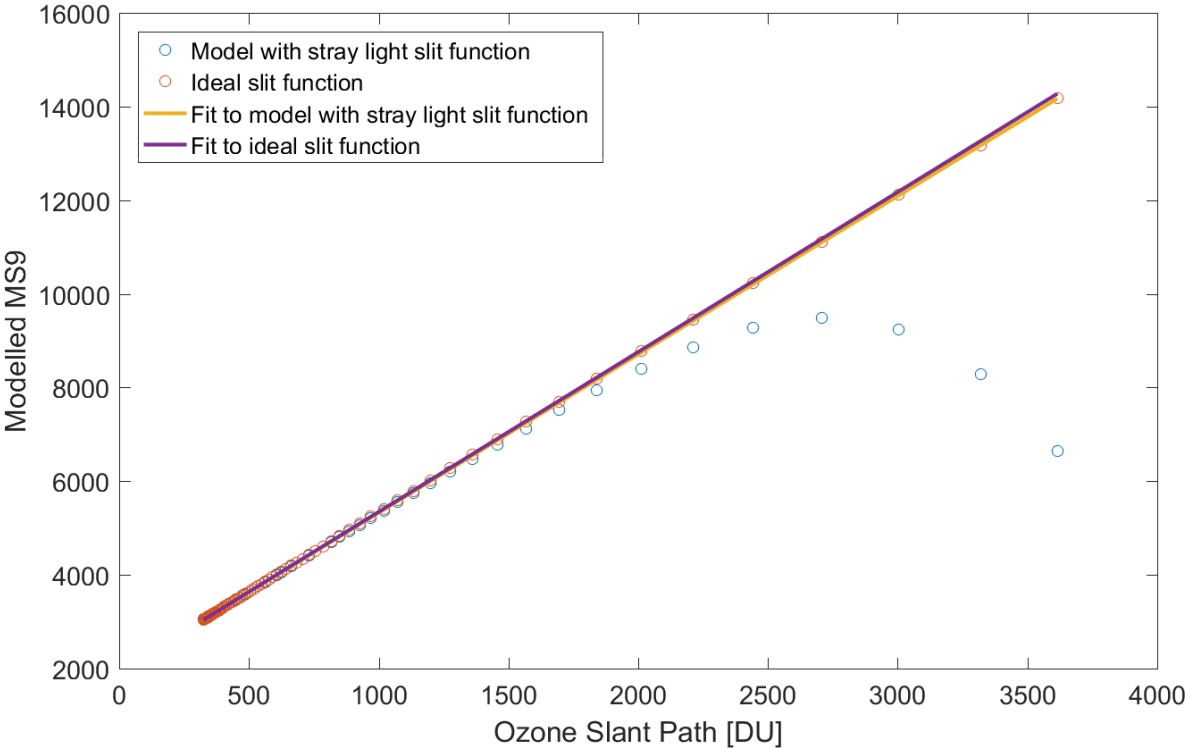

10   **Figure 5: Example of Langley plot fitted to a modelled single Brewer data (see text for the details).**

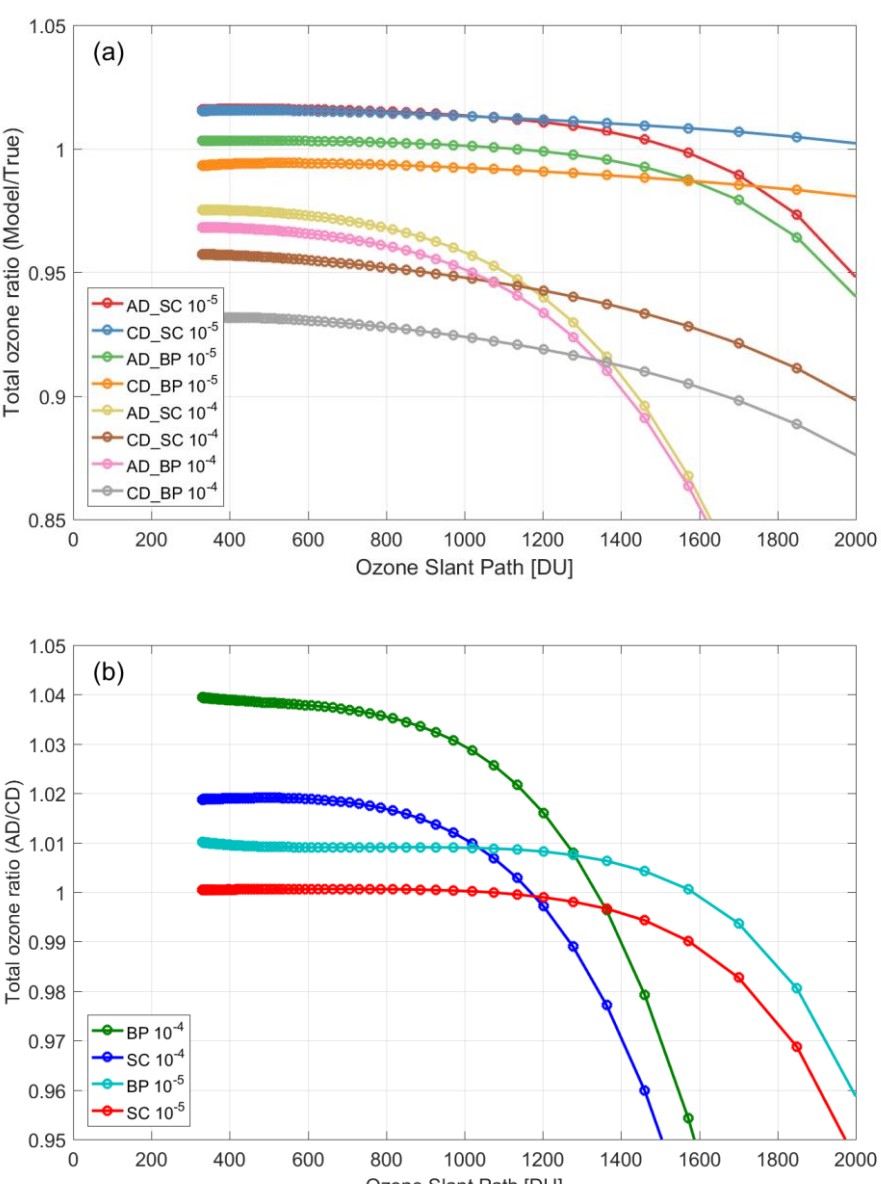

**Figure 6: (a) The ratio of total ozone retrieved from modelled Dobson AD and CD pairs using BP and SC cross-sections with different levels of stray light ($10^{-4}$ and $10^{-5}$) to true ozone as a function of OSP. (b) The ratio of total ozone retrieved from AD to CD wavelength pairs. The adjusted coefficients recommended by WMO are used to derive the total ozone amounts for these models. The ETC values calculated using the Langley method for ideal instruments are used here as well.**

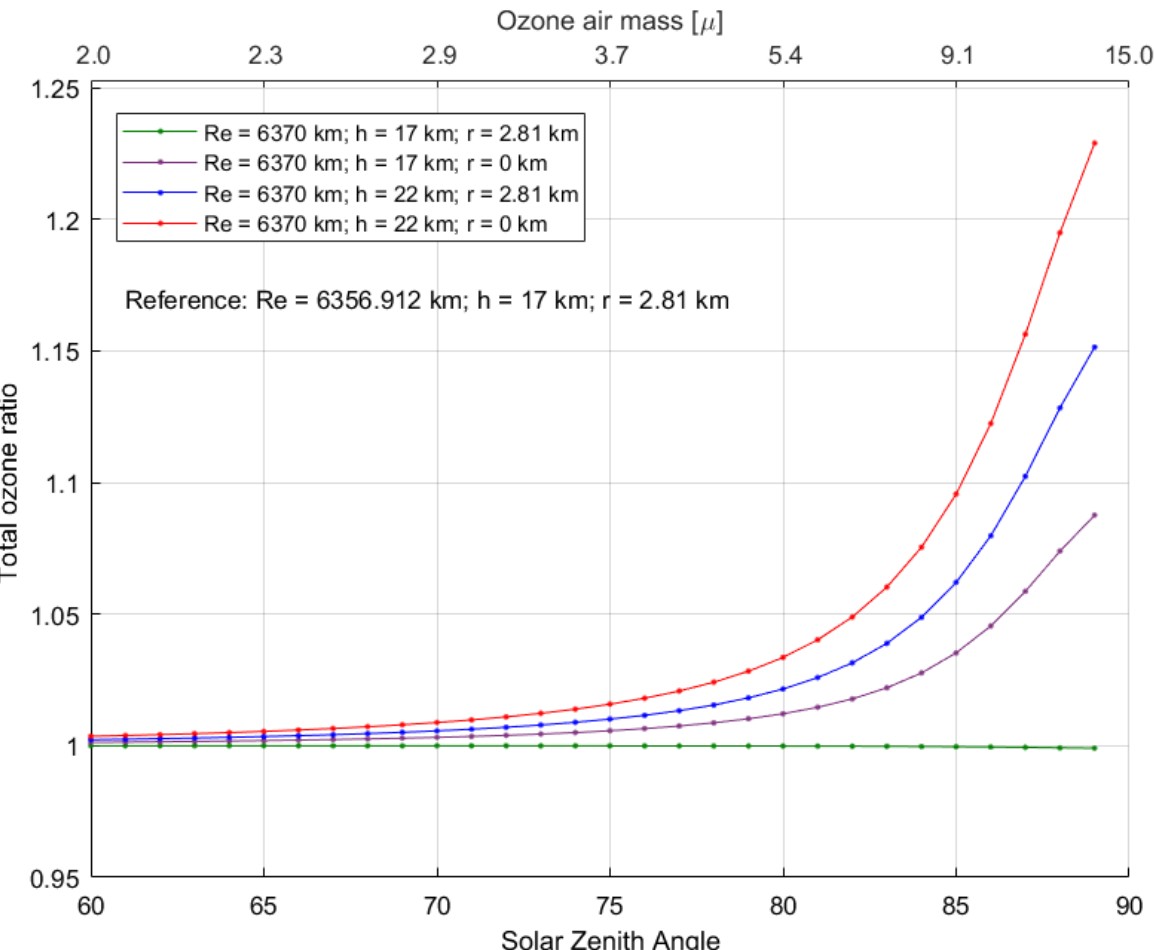

**Figure 7: The ratio of total ozone retrieved from modelled Dobson using air mass factors calculated with a mean value for the radius of the Earth, *Re* = 6370 km as used in the Brewer retrieval, and different values for the altitude of site, *'r'*, and the height of ozone layer, *'h'*, to the ozone values retrieved using air mass factors calculated for the South Pole employing *Re* = 6356.912 km, *r* = 2.81 km, and *h* = 17 km as a reference. Any discrepancy in the air mass is directly reflected in the Dobson and Brewer retrieved total ozone via Eqs. (3) and (9) respectively.**

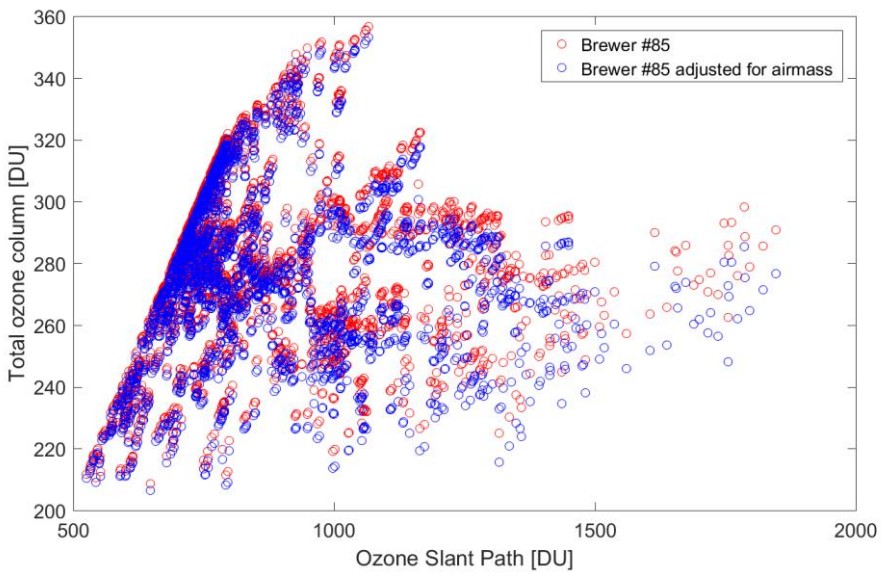

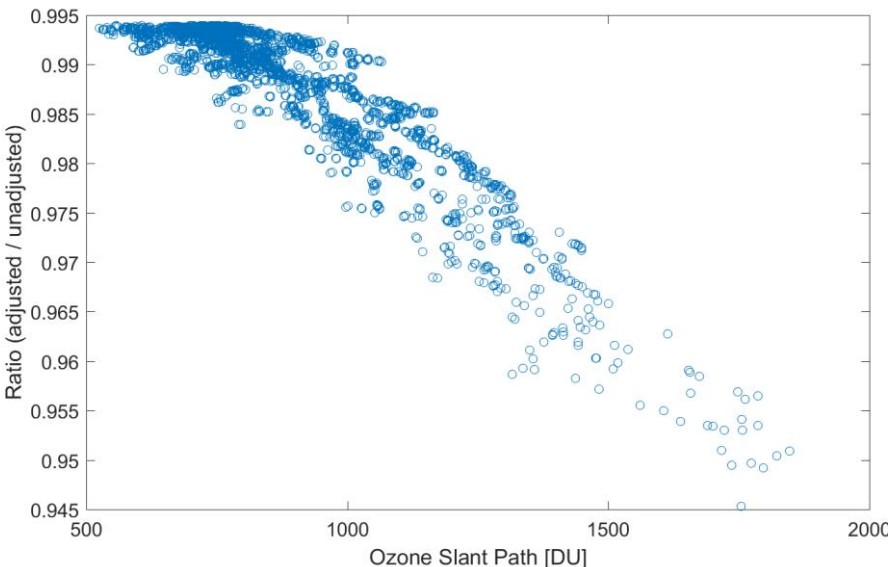

**Figure 8: (Top) Total ozone column values larger than 220 DU measured by double Brewer #085 versus total ozone slant path. Total ozone columns calculated using adjusted air masses are also shown (see the details in the text). (Bottom) Ratio of total ozone column retrieved using adjusted air mass to total ozone retrieved using unadjusted air mass.**

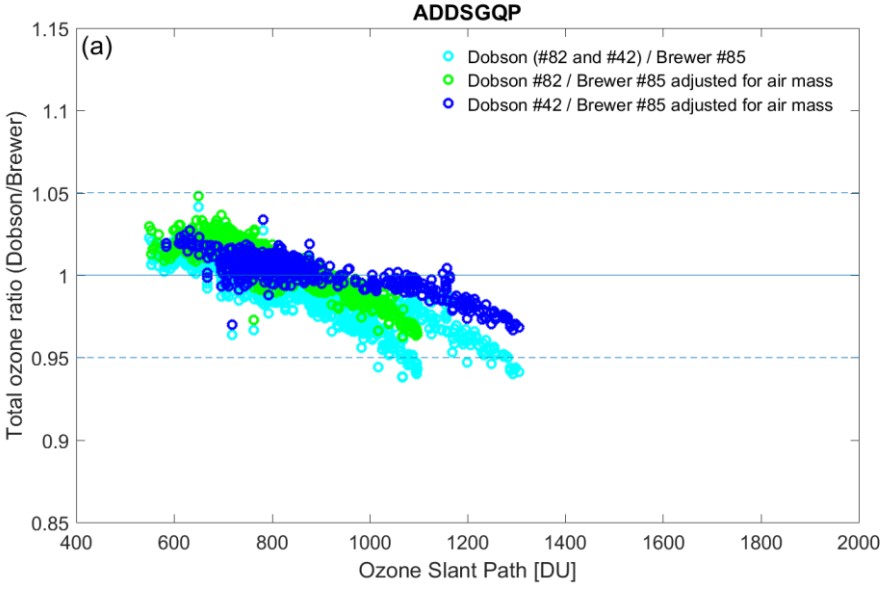

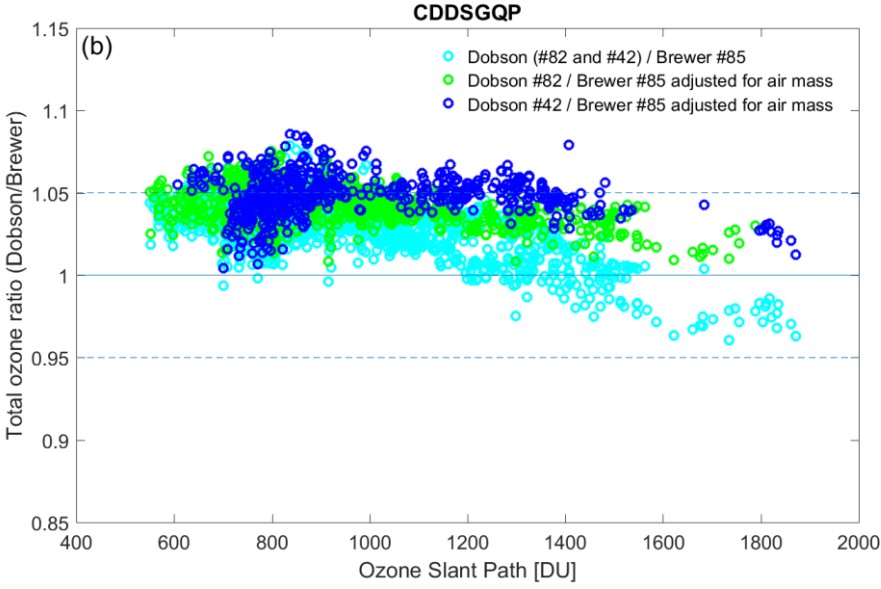

**Figure 9: (a) The ratio of quasi-simultaneous observations (within 5 min) using Dobson (#82 and #42 ) AD wavelengths to double Brewer #085 data at the South Pole for total ozone values larger than 220 DU. (b) Same as (a) using Dobson CD pairs. Brewer air masses have been corrected using the values used for the Dobson measurements for the radius of the Earth, ozone layer height and the altitude of the site. The ratio of all data from three Dobson instruments and Brewer data before adjustment for ozone layer height and station altitude are also depicted. ADDSGQP: AD direct sun measurement using a ground quartz plate, and CDDSGQP: CD direct sun measurement using ground quartz plate.**

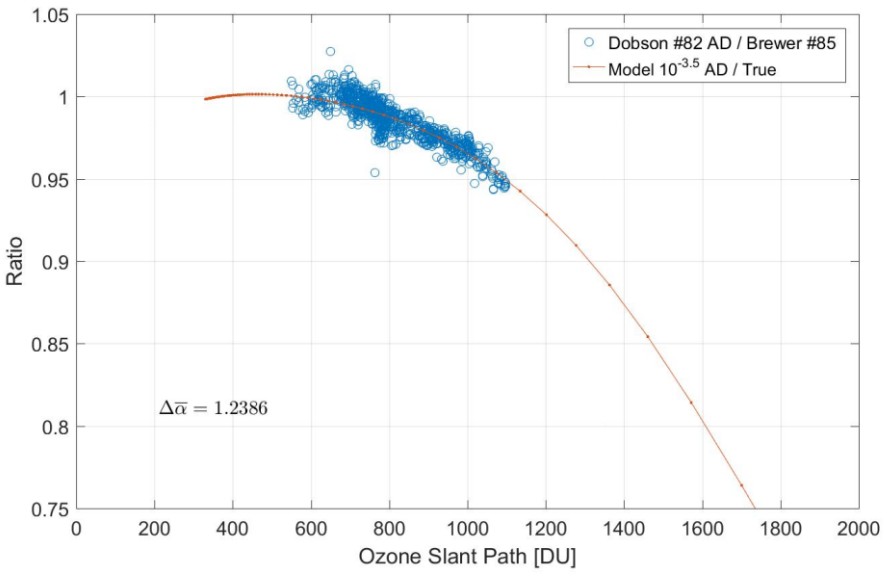

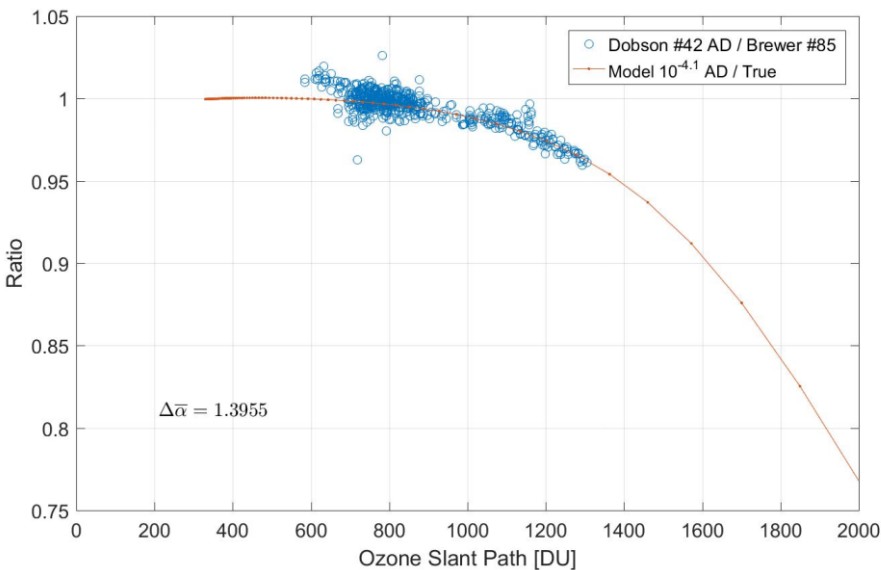

**Figure 10: The ratio of quasi-simultaneous observations of Dobson #82 and #42 AD wavelengths to double Brewer #085 data at the South Pole for total ozone values larger than 220 DU as well as the ratio of the values retrieved from the physical model developed in this study with certain amounts of stray light to the true value assumed in the atmosphere suggesting the level of stray light in each individual Dobson instrument. The ETC values and ozone absorption coefficients are calculated for each model separately using Langley method. Note that the average difference between the Brewer and Dobson data with OSPs less than 800 DU has been used to scale the Dobson data first. Then the model with stray light level that better matches with the Dobson data has been found. The scaling factors used for Dobson #82 and #42 are 1.02 and 1.0076 respectively.**

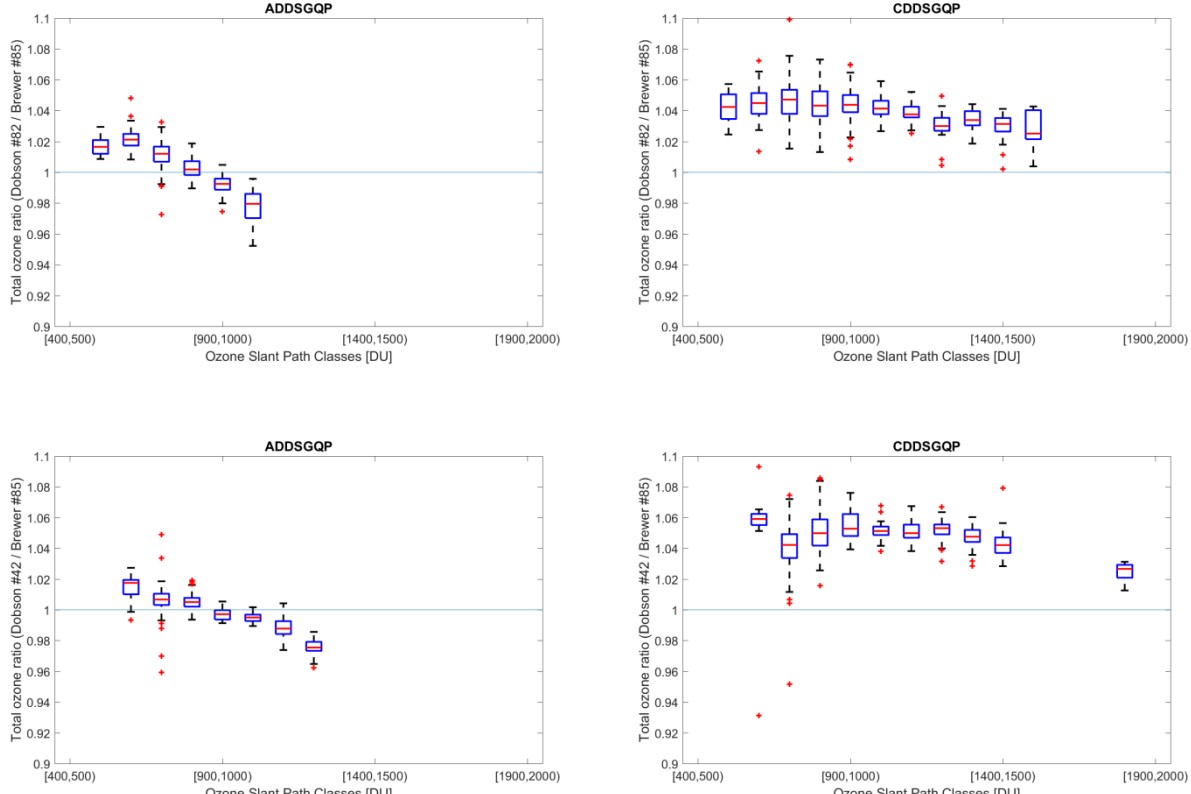

Figure 11: The ratio of quasi-simultaneous direct sun observations (within 5 min) by Dobsons #82 and #42, AD and CD wavelengths to data from double Brewer #085 at the South Pole for total ozone values larger than 220 DU. On each box, the central red line is the median, the edges of the box are the 25th and 75th percentiles, the whiskers extend to the most extreme data points not considered outliers, and outliers are plotted individually. In each bin the values with differences larger than three standard deviations from the mean of the bin have been removed from the calculation. Note that only bins with more than 6 simultaneous measurements are depicted. The title of plots is showing the type of measurements. ADDSGQP: AD direct sun measurement using a ground quartz plate, and CDDSGQP: CD direct sun measurement using ground quartz plate.