# Peer review of "The Effect of Instrumental Stray Light on Brewer and Dobson Total Ozone Measurements"

_Atmospheric Measurement Techniques, 2018_

## Referee Comment (RC1) · Anonymous Referee #1 · 11 May 2018

General Comments

This manuscript calculates the effect of stray light on the ozone absorption cross-sections, and hence the derived values of total ozone, from Dobson and Brewer spectrophotometers. These two instruments have formed the basis of global ground-based measurements of total ozone for many decades and thus this is a very useful issue to address and well within the scope of AMT.

In its current form, however I feel the manuscript suffers from two major defects.

Firstly, I found the logic hard to follow, meaning I was often quite confused about how the different sections related to each other and what the purpose of each really was. Results from different sections didn't seem to even be used in the following sections.

[Figure]

(More details are given in the specific comments). The analysis of measurements at South Pole is only very partially linked back to the model calculations and not at all linked to the lab measurements. The connections and argument need to be made much more explicit.

Secondly, the study seems to have been carried out largely in isolation from work that has been undertaken in the Brewer community over the last five years or so. Some recent references are missing, and others are cited but not sufficiently engaged with.

In particular, I would insist the analysis be re-computed using Serdyuchenko cross-sections. This makes the work relevant to the current day concerns of the community and removes factors that are known to be caused by the use of Bass-Paur (Redondas et al. 2014).

Specific comments Page 1

Line 20 – This needs to be done using Sedyuchenko cross-sections to be relevant and comparable to modern work.

Line 22 – I dispute that you have "evaluated" the error. The discrepancy between Dobson and Brewers as a result of their different assumptions is calculated but nothing here says what the deviation from the true value is.

Line 24 Between 2008 and 2012 – this is quite misleading because you actually only use two distinct periods in 2008 and 2012 (Unless the description on page 11 is wrong?)

Line 25 I can't see that you have shown this at all. You have shown the difference between the Dobson values and the double-Brewer values, but how have you actually attributed this difference to stray light? This is a serious defect that needs to be addressed.

Line 30 I wouldn't say a "similar network" was introduced because of the more limited geographical coverage of Brewers even to today.

Line 15 Refer to Staehlin et al. GAW report

Line 16 I think you need to be specific here – what fraction of the difference can be accounted for?

Line 21 "properly" is not the right word, a lot of work has been done, eg at the RBCCE

Line 4 "large SZA and large TOC" – this is only true in the Northern Hemisphere. It is not true at all in Antarctica, which you use for your comparison. Was South Pole even a good choice for your study?

Line 24 It's fine to do the calculations using Bass-Paur so you can compare them to older work but you also have to do them using Serdyuchenko to be relevant to modern work, eg Redondas et al. 2014, Köhler et al. 2018)

Line 29 the "relevant temperature" – you need to be explicit here – are you using the same temperature for both Dobsons and Brewers? Which is it? Otherwise won't this introduce a difference separate to what you're looking at?

Line 25-27 To be clear, you are not going to use this approximation? (equation 18). You should be explicit.

Line 4 Why do you use theta_0 not just theta?

Line 9 You say "it is important" but don't give any evidence as to why it's important. Evidently the Brewer algorithm doesn't think it's important.

Line 9 You say the "correct" value of the height of the ozone layer but don't show that the Dobson parameterisation is correct. I think you just mean that the Brewer and Dobson methods are different to each other and this will cause a slight difference in derived total ozone.

Line 19 How do you know the stray light in a Dobson is similar to Mk IV and Mk II Brewers? Are you taking this from previous studies? This is one of my major confusions. I don't think you measured it?

Line24 You should mention that He-Cd laser has been used before and give the references (see Pulli et al. 2018)

Line 24-25 You should give at least a very brief description of the experimental set-up. For example, you should explain how you derive a slit-function from a single wavelength laser? What is the sensitivity of your detector? (This is important since you are measuring over such a wide dynamic range).

Line 15 You need to refer to Köhler et al. 2018.

Lines 29-32 I am very confused here about what is what. In Figure 3 the slit functions are curved, not trapezoids. Where did this shape come from?

Line 3 It seems you are not using the approximation in equation 18. Did you use a radiative transfer model?

Lines 8-9 I find this statement completely baffling. What do you mean by "measured slit functions"? What value of stray light are you suggesting WMO use?

Line 15 I would like to see a plot showing what the Langely looked like without and

without the stray light being added to the model

Line 3 It seems to me your results would imply the AD-CD correction should use an expression linear in mu rather than an average across the mu range?

Line 15 You can't say "error" because you don't make any attempt to look at the what the true value is (for example by using South Pole ozonesonde data). You could call it a "discrepancy" between the Dobson and Brewer.

Line 30 Do you mean "February 2008 and December 2014" or is it actually meant to be "February 2008 to December 2014" ?

Line 31 I don't think you can say "corrected" because you don't know that the Dobson value is any more correct than the Brewer value.

Line 9-14 It looks like a linear relation. Couldn't you then calculate the gradient and compare the value with your model? This is what I was expecting you to do to better finish off the study.

Line 24-26 It seems a bit curved. You need to calculate how close the measured values are to what you expect from your model.

Line 28 "Physical model" but four lines later you say "mathematical model".

Line 30 Again, I don't think you have found anything about the "error" – only the difference between the Dobson and Brewer.

Line 32 But does this 25% relate to a realistic value of Dobson stray light?

Page 13 Line 2 "like polar stations" would be better worded as "such as polar stations"

Line 2 This is misleading because the study has considered South Pole data where ozone is very low in spring.

Line 6 You say "stray light also can affect" but isn't this just a different way of expressing the same thing? (ie stray light will affect total ozone, which alternatively you could express as the effect on the absorption coefficients).

Line 10-16 You need to discuss Köhler et al. 2018 here

Line 34 This calculation needs to be with Serdyuchenko cross-sections.

Line 6 "high" should be "higher"

Line 7 "low" should be "lower"

Page 20 Table 3 Are the values for the single or double Brewer?

Page 23 Figure 2 Would this be better on a log scale as in Figure 1?

Page 24 Figure 3 I am very confused about this figure. The shapes are cruved not straight lines. Did you measure these in the lab with the laser?

Page 29 Figure 8 How did you identify the outliers? (If they represent bad data, perhaps you shouldn't plot them?)

REFERENCES

Köhler, U., Nevas, S., McConville, G., Evans, R., Smid, M., Stanek, M., Redondas, A., and Schönenborn, F.: Optical characterisation of three reference Dobsons in the ATMOZ Project – verification of G. M. B. Dobson's original specifications, Atmos. Meas. Tech., 11, 1989-1999, https://doi.org/10.5194/amt-11-1989-2018, 2018.

Koukouli, M. E., Zara, M., Lerot, C., Fragkos, K., Balis, D., van Roozendael, M., Allart, M. A. F., and van der A, R. J.: The impact of the ozone effective temperature on satellite validation using the Dobson spectrophotometer network, Atmos. Meas. Tech., 9, 2055-2065, https://doi.org/10.5194/amt-9-2055-2016, 2016.

T. Pulli, T. Karppinen, S. Nevas, P. Kärhä, K. Lakkala, J. M. Karhu, M. Sildoja, A.

Vaskuri, M. Shpak, F. Manoocheri, L. Doppler, S. Gross, J. Mes & E. Ikonen (2018) Out-of-Range Stray Light Characterization of Single-Monochromator Brewer Spectrophotometers, Atmosphere-Ocean, 56:1,1-11, DOI: 10.1080/07055900.2017.1419335

Staehelin, J., Kerr, J., Evans, K., and Vanicek, R.: Comparison Of Total Ozone Measurements of Dobson and Brewer spectrophotometers and Recommended Transfer Functions, WMO TD No. 1147, World Meteorological Organization, Global AtmosphereWatch, No. 149, available at: http://www.wmo.ch/web/arep/reports/gaw149.pdf., 2003.

Stübi, R., H. Schill, J. Klausen, L. Vuilleumier, and D. C. Ruffieux (2017), Reproducibility of total ozone column monitoring by the Arosa Brewer spectrophotometer triad, J Geophys. Res. Atmos., 122, 4735–4745, doi:10.1002/2016JD025735.

---

## Referee Comment (RC2) · A. Redondas (Referee) · 19 May 2018

**1   General comments**

The article is interesting as the effect of the stray light on the ozone cross section calculation were not studied on the past. I have a few comments which I would the authors to answer, pending those I support the publication of the manuscript.

The principal comment i repeat from my first evaluation is why they don't use the ( Serdyuchenko 2014) cross section in his calculations , when is now the recommended ozone cross section for Brewer and Dobson. Moreover some of the discussions of the paper like the AD/CD ozone difference in the Dobson measurements and the

Brewer/Dobson differences are also affected with the change of cross section (Redondas 2014). The discussion (Section 3.1 ) is still difficult to follow especially the Dobson section (see specific comments).

The second point to mention is the ETC calculation in section 3.2, is not clear how is calculated, in particular how is related from Chance and Kurucz (table 2).

As suggested by the referee Julian Groebner on the discussion on the paper also part of this special number ( Redondas 2018). The difference between the use of trapezoid slit and a triangular slit is about 0.7% in a double brewer. I think is usefull to include this case in your calculation.

The error introduced by the assumption of the fixed air-mass is also showed but could be more illustrative to show the difference in ozone rather than in airmass. The effect on the Dobson record at South Pole was also studied by (?) a comparison with his results could be also illustrative.

**2  Specific comments**

Page 2, 30: A basic description of the method is worthwhile, the method is based on the characterization of the instrument and need both the spectral response and the Laser measurements of the slit rather than the dispersion information. A comparison between the Kiedrom/Karppinen model ant this work will be illustrative.

Page 3, 7: A reference of the false positive trend due different stray light is advisable.

Page 4: There is no explanation of the calibration of the Dobson as is done with the Brewer

Page 5: There is some confusion on the nomenclature of the formulas: please unify B or ETC, F or I.

Page 5,29 : A reference of the application of the Barnes correction to the Brewer network will be advisable.

Page 6,1 : There are several files available at IGACO, which ones are used in this study?.

Page 6,20 Could clarify the relation between equations 11, 15,17 and 18. (se allso 9,1 ).

Page 7.15: A mention of a other sources of stray light could be mention, see for example Josefsson and discused.

Page 9,1: There is a confusing use of $\overline{\alpha_i}$ vs $\alpha_i^{approx}$ where are talking about $\Delta\alpha$. The same issue for table 3.

Page 9,5: Is surprising that the calculation of the operational values agree with yours calculations. In this work you are using a different cross section temperature, brewer uses -45 C but you are using -46.3 C (Table 2). The same nominal wavelengths (Table 1) for both brewers whereas brewer operative wavelengths are slightly different for every instrument, and the same FWHM for all the slits. Can be also useful to have the brewer ozone absorption coefficient for for every wavelength ( $\overline{\alpha_i}$ vs $\alpha_i^{approx}$) and not only the effective $\Delta\alpha$.

Page 10,5: An explanation why the calibration method reduces the the discrepancy to 0.7% independent of the level of stray light of the instruments and can be illustrated for example in figure 5.

Page 10,15: Please describe the calculation of the ETC, how is compared with the calculation of Kiedrom and (Karppinen 2015).

Page 10.25: Consider also to discuss the case of early spring at high latitudes, with low sun and high ozone content.

Page 11.15: Consider to plot corrected /uncorrected South Pole Brewer to illustrate the

error due air mass calculation.

Page 12: A better description of the data-set might be provided or referenced: number of simultaneous measurement, if the data are available at WOUDC/NDACC databases, the QA/QC results of calibrations and how stable are in time the comparison between Dodson instruments, and brewer-dobson will help to interpret the comparison.

Page 12.15: Concerning the analysis , the intervals with a reduced number of observations should be removed, this discard for example most of the Dobson #80 observation for high ozone slant column. Consider to use the same order of the Dobson instruments in plots and enumerations.

Page 13,5 : In the conclusions refers that you are using the measured slit but in reality the central part of the slit is not measured, and are also model as trapezoid.

**3  References**

Bernhard, G., Evans, R. D., Labow, G. J., and Oltmans, S. J.: Bias in Dobson total ozone measurements at high latitudes due to approximationsin calculations of ozone absorption coefficients and air mass, 110, https://doi.org/doi:10.1029/2004JD005559,.

Josefsson, W.: Focused sun observations using a Brewer ozone spectrophotometer, 97, 15 813–15 817, https://doi.org/10.1029/92JD01030,https://agupubs.onlinelibrary.wiley.com/doi/abs/10.1029/92JD01030.5

Karppinen, T., Redondas, A., García, R. D., Lakkala, K., McElroy, C., and Kyrö, E.: Compensating for the effects of stray light in single-monochromator Brewer spectrophotometer ozone retrieval, Atmosphere-Ocean, 53, 66–73, 2015.

Redondas, A., Evans, R., Stuebi, R., Köhler, U., and Weber, M.: Evaluation of the use of five laboratory-determined ozone absorption crosssections in Brewer and Dobson

retrieval algorithms, Atmospheric Chemistry and Physics, 14, 1635–1648, 2014.

Redondas, A., Nevas, S., Berjón, A., Sildoja, M.-M., León-Luis, S. F., Carreño, V., and Santana, D.: Wavelength calibration of Brewer10spectrophotometer using a tuneable pulsed laser and implications to the Brewer ozone retrieval, Atmos. Meas. Tech. Discuss., 2018, 1–16,https://doi.org/10.5194/amt-2017-459, https://www.atmos-meas-tech-discuss.net/amt-2017-459/, 2018.

Serdyuchenko, A., Gorshelev, V., Weber, M., Chehade, W., and Burrows, J.: High spectral resolution ozone absorption cross-sections–Part2: Temperature dependence, Atmospheric Measurement Techniques, 7, 625–636, 2014.

Please also note the supplement to this comment:
https://www.atmos-meas-tech-discuss.net/amt-2018-2/amt-2018-2-RC2-supplement.pdf

––––––––––––––––––––––––––

---

## Author Comment (AC1) · 29 Jun 2018

**Omid Moeini et al.** (omidmns@yorku.ca)

The authors would like to thank the reviewer for the thorough and useful review which we have gratefully considered in improving the paper.

**A. Redondas (Referee)  aredondasm@aemet.es**

**General comments**

The article is interesting as the effect of the stray light on the ozone cross section calculation were not studied on the past. I have a few comments which I would the authors to answer, pending those I support the publication of the manuscript. The principal comment i repeat from my first evaluation is why they don't use the (Serdyuchenko 2014) cross section in his calculations, when is now the recommended ozone cross section for Brewer and Dobson. Moreover, some of the discussions of the paper like the AD/CD ozone difference in the Dobson measurements and the Brewer/Dobson differences are also affected with the change of cross section (Redondas 2014). The discussion (Section 3.1) is still difficult to follow especially the Dobson section (see specific comments).

The authors deliberately avoid the use of the Serdyuchenko (2014) cross-sections as it is not the intent of this study to presume to set the cross-section values to be used by the community to measure ozone, but only to provide some additional information about the impact of stray light on the measurements. Changing the cross-sections would not change the significance of this study. The cross-sections can be considered as a variable and the nature of the processes would remain the same. The paper is intended to show the connection between the physics of the instruments and the impact of stray light on the measurements. The first priority in using the results of this study is the provision of an algorithm for correcting the extant ozone historical record (described a paper currently in review) – particularly results measured at large slant column ozone amounts (e.g.: particularly at high latitudes in spring and fall). The inclusion of multiple results would obscure the basic intent of the paper and possibly create confusion in the community. Calculating the ozone absorption coefficients with new cross-sections is beyond the

scope of this article and subject of another study which should be supported by WMO. To the best of our knowledge no new data have been submitted to the WOUDC using new cross-sections nor have the historical data been so corrected.

About the Dobson AD-CD difference: This work shows that higher levels of stray light lead to larger differences between AD and CD measurements. This fact is the same for all cross-sections. Redondas et al. (2014) have shown that for the ideal slit functions without stray light, the difference is somewhat lower for the Serdyuchenko (2014) cross-sections as compared with the results using other cross-sections.

The second point to mention is the ETC calculation in section 3.2, is not clear how is calculated, in particular how is related from Chance and Kurucz (table 2). As suggested by the referee Julian Groebner on the discussion on the paper also part of this special number (Redondas et al. 2018). The difference between the use of trapezoid slit and a triangular slit is about 0.7% in a double brewer. I think is usefull to include this case in your calculation.

The ETC calculation and its relation to solar spectrum are discussed and added to the manuscript (see response to specific comments).

The authors believe that repeating all this work with different minor variations (replacing trapezoid slit with triangular slit) and adding all those numbers to the manuscript actually obscures the point of the paper.

The error introduced by the assumption of the fixed air-mass is also showed but could be more illustrative to show the difference in ozone rather than in airmass. The effect on the Dobson record at South Pole was also studied by (Bernhard et al. 2005) a comparison with his results could be also illustrative.

The plot that shows the difference in airmass is replaced with a plot showing the difference in ozone.

Generally, Bernhard et al. (2005) has stated that the effective ozone absorption coefficients, $\Delta\bar{\alpha}_{AD}$ and $\Delta\bar{\alpha}_{CD}$, are smaller at large ozone slant path which is the same result as from the current analysis. They have compared the Dobson data with TOMS and SUV-100 data. To compare those results with this analysis, information about the TOMS and SUV-100 instruments would be required. This is beyond the scope of this article and could be a subject for another study.

**Specific comments**

Page 2, 30: A basic description of the method is worthwhile, the method is based on the characterization of the instrument and need both the spectral response and the Laser measurements of the slit rather than the dispersion information. A comparison between the Kiedrom/Karppinen model ant this work will be illustrative.

Kiedrom/Karppinen have tried to correct the stray light effect at large ozone slant paths for a single Brewer by changing the weighting coefficients. Here, two types of Brewers and a Dobson instrument are modeled to show the connection between the physics of the instruments and the impact of stray light on the measurements.

Page 3, 7: A reference of the false positive trend due different stray light is advisable.

This is difficult quantify in a really useful way until a reanalysis of an appropriate data set is done. It is considered beyond the scope of the paper and work that needs to be done.

Page 4: There is no explanation of the calibration of the Dobson as is done with the Brewer

The comment is not clear.

The Dobson calibration procedure is described in Evans and Komhyr (2008). Generally, the Dobsons are adjusted to make the slit functions as similar as possible and then only an extraterrestrial value is transferred from a reference instrument traceable to the World Standard Dobson.

Page 5: There is some confusion on the nomenclature of the formulas: please unify B or ETC, F or I.

Revised. "I" and "ETC" are used.

Page 5,29 : A reference of the application of the Barnes correction to the Brewer network will be advisable.

Revised. "the Brewer" is removed from the statement.

Page 6,1: There are several files available at IGACO, which ones are used in this study?.

The following statement has been added to the manuscript:

"For this study the quadratic coefficients on the file 'Bp.par' are used for BP cross-sections and the Liu et al. (2007) quadratic approximation, which excludes -273º K data from the quadratic temperature dependence fitting, are used for BDM cross-sections."

Page 6,20: Could clarify the relation between equations 11, 15,17 and 18. (see also 9,1).

The ozone absorption coefficients $\alpha(\lambda_i)$ are calculated from ozone absorption cross-sections and vertical profiles of ozone and temperature employing equation 15. Due to the finite bandpass of the Brewer and Dobson's slit functions, the effective ozone absorption coefficients ($\bar{\alpha}(\lambda_i)$) must be calculated either using equation 17 or 18. Equation 17 considers the solar spectrum whereas equation 18 is the simplest approach that can be used to calculate the effective ozone absorption coefficients. The effective ozone absorption coefficients must be used in equation (11).

To prevent any confusion, $\alpha(\lambda_i)$ and $\bar{\alpha}(\lambda_i)$ are used instead of $\alpha_i$ and $\bar{\alpha}_i$ in all equations.

$\alpha(\lambda_i)$ is replaced with $\bar{\alpha}(\lambda_i)$ in equation (11) and $\Delta\alpha$ is replaced with $\Delta\bar{\alpha}$ in equations (3), (8), (9), and (11) to make clear that the effective differential ozone absorption coefficients must be calculated and used in Dobson and Brewer retrievals.

Page 7.15: A mention of a other sources of stray light could be mention, see for example Josefsson and discused.

Revised. The reference and following statement have been added to the manuscript:

"radiation scattered from the atmosphere within field of view of the instrument can also contribute a stray light effect (Josefsson, 1992)."

Josefsson, A. P., (1992), Focused Sun Observation Using a Brewer Ozone Spectrophotometer, *J. Geophys. Res.*, *97*(D14), 15813–15817.

Page 9,1: There is a confusing use of $\alpha_i$ vs $\alpha_i^{approx}$ where are talking about $\Delta\alpha$. The same issue for table 3.

Revised.

Page 9,5: Is surprising that the calculation of the operational values agree with yours calculations. In this work you are using a different cross section temperature, brewer uses -45 C but you are using -46.3 C (Table 2). The same nominal wavelengths (Table 1) for both brewers whereas brewer operative wavelengths are slightly different for every instrument, and the same FWHM for all the slits. Can be also useful to have the brewer ozone absorption coefficient for for every wavelength ($\alpha_i$ vs $\alpha_i^{approx}$ ) and not only the effective $\Delta\alpha$.

Apparently it is a coincidence that those numbers are matched. To validate the procedure, the calculations are repeated for nominal wavelengths of 310.05, 313.50, 316.80 and 320.00 nm with FWHMs of 0.359, 0.555, 0.545 and 0.538 and BP cross-sections at -45 °C and compared with the value calculated by Redondas et al. (2014) using the same cross-sections for a nominal Brewer. There is a difference of 0.06 % between this value (0.3365) and the value calculated by Redondas et al. (2014) (0.3367) using IGQ4 cross-sections for a nominal Brewer (Table 6) which is identical with our double Brewer in terms of slit functions.

Also, the Brewer ozone absorption coefficients for every wavelength are calculated and reported as suggested by the referee.

The Brewer part of the discussion (section 3.1) is revised and following statements and tables are added to the manuscript:

"To validate the calculations, the $\Delta\bar{\alpha}^{apx}$ is calculated for the double Brewer using an ideal trapezoid slit function and BP cross-sections at -45 °C without the Barnes (1987) correction. Redondas et al. (2014) have calculated the ozone absorption coefficients for the nominal Brewer which is identical in terms of slit functions, nominal wavelengths and slit FWHMs with the double Brewer of this work using ideal trapezoid slit functions. The IGQ4 cross-sections used in Redondas et al. (2014) are the same as the BP cross-sections employed at this work. The value 0.3367 calculated using IGQ4 cross-sections at -45 °C (Redondas et al. Table 6) has a difference of 0.06 % with the value 0.3365 calculated here with the same cross-sections at the same temperature (-45 °C) (Table 3).

To be consistent with Dobson calculations, the BP cross-sections with Barnes (1987) correction and at -46.3 °C are used for calculation of $\bar{\alpha}_i$ and $\bar{\alpha}_i^{apx}$ presented in tables 4 and 5 for the single and double Brewers.

The contribution of stray light in determining the ozone absorption coefficients can be seen from comparing the $\Delta\bar{\alpha}$ calculated using ideal slit functions (without stray light) with the values ($\Delta\bar{\alpha}$) calculated using modeled slit functions (including stray light). For the single Brewer the results show a 0.7 % difference (modeled slit functions including stray light are less than that of the ideal slit functions) while for the double Brewer the difference is less than 0.01 %.

Comparing $\Delta\bar{\alpha}$ with $\Delta\bar{\alpha}^{apx}$ for both Brewers (single and double using ideal and modeled slit functions) shows a minimum difference of 0.7 % ($\Delta\bar{\alpha}$ higher than $\Delta\bar{\alpha}^{apx}$) for the double Brewer

with ideal trapezoid slit functions and a maximum of 0.9 % for single Brewers with ideal triangle slit functions, indicating the role of the solar spectrum in calculating the ozone absorption coefficients.

Table 3: Ozone absorption coefficients calculated here and the value calculated by Redondas et al. (2014)

| Wavelength (nm) | FWHM (nm) | $\bar{\alpha}_i^{apx}$ (atm cm$^{-1}$) calculated for Double Brewer using ideal slits and BP cross-sections at -45 $^o$C without Barnes (1987) correction | From Redondas (2014) Table 6; effective ozone absorption coefficient calculated using IQG4 B&P cross-sections |
|---|---|---|---|
| | | Ideal (trapezoid) | Ideal (trapezoid) |
| 310.05 | 0.539 | 1.0044 | |
| 313.50 | 0.555 | 0.6793 | |
| 316.80 | 0.545 | 0.3760 | |
| 320.00 | 0.538 | 0.2935 | |
| $\Delta\bar{\alpha}^{apx}$ | | 0.3365 | 0.3367 |

Table 4: Single Brewer ozone absorption coefficients

| Wavelength (nm) | FWHM (nm) | Ideal | | Model (with Stray light) | |
|---|---|---|---|---|---|
| | | $\bar{\alpha}_i^{apx}$ | $\bar{\alpha}_i$ | $\bar{\alpha}_i^{apx}$ | $\bar{\alpha}_i$ |
| 310.05 | 0.539 | 1.0087 | 1.0127 | 1.0141 | 1.0102 |
| 313.50 | 0.555 | 0.6824 | 0.6842 | 0.6828 | 0.6833 |
| 316.80 | 0.545 | 0.3774 | 0.3789 | 0.3768 | 0.3789 |
| 320.00 | 0.538 | 0.2944 | 0.2962 | 0.2923 | 0.2959 |
| $\Delta\bar{\alpha}^{apx}/\Delta\bar{\alpha}$ | | 0.3377 | 0.3406 | 0.3407 | 0.3380 |

*BP cross-sections at -46.3 with Barnes (1987) correction.

Table 5: Double Brewer ozone absorption coefficients

| Wavelength (nm) | FWHM (nm) | Ideal | | Model (with Stray light) | |
|---|---|---|---|---|---|
| | | $\bar{\alpha}_i^{apx}$ | $\bar{\alpha}_i$ | $\bar{\alpha}_i^{apx}$ | $\bar{\alpha}_i$ |
| 310.05 | 0.539 | 1.0087 | 1.0127 | 1.0089 | 1.0126 |
| 313.5 | 0.555 | 0.6824 | 0.6842 | 0.6826 | 0.6841 |
| 316.8 | 0.545 | 0.3773 | 0.3789 | 0.3776 | 0.3789 |
| 320 | 0.538 | 0.2947 | 0.2962 | 0.2950 | 0.2962 |
| $\Delta\bar{\alpha}^{apx}/\Delta\bar{\alpha}$ | | 0.3384 | 0.3406 | 0.3384 | 0.3405 |

*BP cross-sections at -46.3 with Barnes (1987) correction.

Page 10,5: An explanation why the calibration method reduces the the discrepancy to 0.7% independent of the level of stray light of the instruments and can be illustrated for example in figure 5.

The following statement has been added:

"In the Dobson AD pair calibration, scale factors are calculated for different ranges of airmass. The data from the instrument being calibrated are scaled to the data from the reference instrument. Then the CD pair data of the calibrated instrument are scaled to its AD pair data."

Page 10,15: Please describe the calculation of the ETC, how is compared with the calculation of Kiedrom and (Karppinen 2015).

The following statement and plot have been added to the manuscript:

"To calculate the ETC, the instrument absorption function using the solar spectrum (Chance and Kurucz, 2010), Eqs. (1) and (2) and the retrieval algorithm of the Brewer (or Dobson) for an assumed constant amount of ozone (325 DU in this study) is calculated and plotted as a function of ozone slant path. The best fit to the data with airmass less than 2 (less than 3 for the Dobson instruments) is found and extrapolated to zero airmass. Figure 5 shows the best fit to the single Brewer data:

[Figure]

**Figure 5: Example of Langley plot fitted to a modeled single Brewer data**

For the single Brewer the ETC is calculated as 1945.4 for a modeled trapezoid slit function with stray light which is comparable with 2020 as calculated by Kiedrom et al. (2008) noting the slight differences in the slit functions and solar spectrum. Karppinen et al. (2015) have reported 3218 for ETC value for slit functions with stray light. However, they used LibRadtran 1.6-beta radiative transfer model to scale their data to be matched with real data."

Page 10.25: Consider also to discuss the case of early spring at high latitudes, with low sun and high ozone content.

This is a good suggestion but beyond the scope of this study. We chose the South Pole as it is important for the detection of ozone recovery and is being used for trend analysis and satellite validations. With ozone recovery in future more data collected at large ozone slant paths would be available which may cause an error in the trend analysis or satellite validation due to the effect of stray light.

Page 11.15: Consider to plot corrected /uncorrected South Pole Brewer to illustrate the error due air mass calculation.

The following plot has been added to the manuscript.

[Figure]

Page 12: A better description of the data-set might be provided or referenced: number of simultaneous measurement, if the data are available at WOUDC/NDACC databases, the QA/QC results of calibrations and how stable are in time the comparison between Dodson instruments, and brewer-dobson will help to interpret the comparison.

The following statements have been added to the manuscript:

"The Brewer data for the South Pole site are available at the WOUDC website. Due to the logistic difficulties Brewer #085 was not replaced or calibrated until 2016.

The Dobson data used for this study are freely available at: ftp://aftp.cmdl.noaa.gov/user/evans/York_Omid/. For this study all direct sun Dobson measurements are used while only one measurement representative of the day is reported to the NDACC or WOUDC. A complete description of the South Pole dataset is provided by Evans et al. (2017). The reprocessed data using WinDobson software as described in Evans at al. (2017) are used for the analysis here. Generally, the Dobson instrument at the South Pole site is replaced with a calibrated instrument every four years. The instrument replaced is calibrated against the reference Dobson #083 and the calibration results are used to adjust and post-process the last four years of data collected at the South Pole. The calibration procedure can be found at Evans and Komhyr (2008) and the major calibration or instrument changes regarding the South Pole dataset can be seen in Fig. 5 of Evans et al. (2017)."

The simultaneous measurements available are summarized in a table and added to the manuscript:

"Table 7: The number of simultaneous measurements in each bin

|  | Dobson #82 | | Dobson #42 | | Dobson #80 | |
| --- | --- | --- | --- | --- | --- | --- |
| Bins (OSP) | AD | CD | AD | CD | AD | CD |
| [400 500) | 39 | 33 | 0 | 0 | 45 | 41 |
| [500 600) | 171 | 143 | 7 | 0 | 63 | 63 |
| [600 700) | 172 | 113 | 101 | 70 | 57 | 72 |
| [700 800) | 439 | 313 | 258 | 179 | 11 | 8 |
| [800 900) | 174 | 235 | 153 | 178 | 5 | 6 |
| [900 1000) | 155 | 120 | 30 | 54 | 0 | 1 |
| [1000 1100) | 96 | 125 | 57 | 28 | 0 | 0 |
| [1100 1200) | 4 | 50 | 46 | 67 | 7 | 4 |
| [1200 1300) | 0 | 41 | 36 | 46 | 0 | 2 |
| [1300 1400) | 0 | 43 | 4 | 49 | 3 | 2 |
| [1400 1500) | 0 | 36 | 0 | 19 | 0 | 1 |
| [1500 1600) | 0 | 19 | 0 | 4 | 0 | 0 |
| [1600 1700) | 0 | 6 | 0 | 1 | 0 | 0 |
| [1700 1800) | 0 | 9 | 0 | 1 | 0 | 0 |
| [1800 1900) | 0 | 0 | 0 | 10 | 0 | 0 |
| [1900 2000) | 0 | 0 | 0 | 0 | 0 | 0 |
| Total | 1250 | 1286 | 692 | 706 | 191 | 200 |

 Concerning the analysis , the intervals with a reduced number of observations should be removed, this discard for example most of the Dobson #80 observation for high ozone slant column. Consider to use the same order of the Dobson instruments in plots and enumerations.

The boxes with less than 6 coincident measurements in each bin are removed from the plots.

[Figure]

Page 13,5 : In the conclusions refers that you are using the measured slit but in reality the central part of the slit is not measured, and are also model as trapezoid.

Revised.

**References**

Bernhard, G., Evans, R. D., Labow, G. J., and Oltmans, S. J.: Bias in Dobson total ozone measurements at high latitudes due to approximationsin calculations of ozone absorption coefficients and air mass, 110, https://doi.org/doi:10.1029/2004JD005559,.

Josefsson, W.: Focused sun observations using a Brewer ozone spectrophotometer, 97, 15 813–15 817, https://doi.org/10.1029/92JD01030,https://agupubs.onlinelibrary.wiley.com/doi/abs/10.1029/92JD01030.5

Karppinen, T., Redondas, A., García, R. D., Lakkala, K., McElroy, C., and Kyrö, E.: Compensating for the effects of stray light in single-monochromator Brewer spectrophotometer ozone retrieval, Atmosphere-Ocean, 53, 66–73, 2015.

Redondas, A., Evans, R., Stuebi, R., Köhler, U., and Weber, M.: Evaluation of the use of five laboratory-determined ozone absorption crosssections in Brewer and Dobson retrieval algorithms, Atmospheric Chemistry and Physics, 14, 1635–1648, 2014.

Redondas, A., Nevas, S., Berjón, A., Sildoja, M.-M., León-Luis, S. F., Carreño, V., and Santana, D.: Wavelength calibration of Brewer10spectrophotometer using a tuneable pulsed laser and implications to the Brewer ozone retrieval, Atmos. Meas. Tech. Discuss., 2018, 1–16,https://doi.org/10.5194/amt-2017-459, https://www.atmos-meastech-discuss.net/amt-2017-459/, 2018.

Serdyuchenko, A., Gorshelev, V., Weber, M., Chehade, W., and Burrows, J.: High spectral resolution ozone absorption cross-sections–Part2: Temperature dependence, Atmospheric Measurement Techniques, 7, 625–636, 2014.

Evans, R. D., Petropavlovskikh, I., Mcclure-begley, A., Mcconville, G., and Quincy, D.: Technical note : The US Dobson station network data record prior to 2015 , re-evaluation of NDACC and WOUDC archived records with WinDobson processing software, , 12051–12070, 2017.

Evans, R., and Komhyr, W.: Operations Handbook - Ozone Observations with a Dobson Spectrophotometer, WMO/GAW Report No. 183, World Meteorological Organization, Geneva, Switzerland, 2008.

---

## Author Comment (AC2) · 29 Jun 2018

**Omid Moeini et al.** (omidmns@yorku.ca)

The authors would like to thank the reviewer for the thorough and useful review which we have gratefully considered in improving the paper.

**Anonymous Referee #1**

**General Comments**

This manuscript calculates the effect of stray light on the ozone absorption cross-sections, and hence the derived values of total ozone, from Dobson and Brewer spectrophotometers.

These two instruments have formed the basis of global ground-based measurements of total ozone for many decades and thus this is a very useful issue to address and well within the scope of AMT.

In its current form, however I feel the manuscript suffers from two major defects.

Firstly, I found the logic hard to follow, meaning I was often quite confused about how the different sections related to each other and what the purpose of each really was.

Results from different sections didn't seem to even be used in the following sections.

(More details are given in the specific comments).

The analysis of measurements at South Pole is only very partially linked back to the model calculations and not at all linked to the lab measurements. The connections and argument need to be made much more explicit.

The discussion (section 3) is revised to be explicit and clear and link different parts of the article. The model developed in this study is used to estimate the stray light level within each Dobson instrument located at South Pole (please see the response to the specific comments and also the response to A. Redondas (page 9,5)).

Secondly, the study seems to have been carried out largely in isolation from work that has been undertaken in the Brewer community over the last five years or so. Some recent references are missing, and others are cited but not sufficiently engaged with.

The recent studies have been discussed and the references have been added to the manuscript (please see the response to the specific comments).

In particular, I would insist the analysis be re-computed using Serdyuchenko cross-sections. This makes the work relevant to the current day concerns of the community and removes factors that are known to be caused by the use of Bass-Paur (Redondas et al. 2014).

The authors deliberately avoid the use of the Serdyuchenko (2014) cross-sections as it is not the intent of this study to presume to set the cross-section values to be used by the community to measure ozone, but only to provide some additional information about the impact of stray light on the measurements. Changing the cross-sections would not change the significance of this study. The cross-sections can be considered as a variable and the nature of the processes would remain the same. The paper is intended to show the connection between the physics of the instruments and the impact of stray light on the measurements. The first priority in using the results of this study is the provision of an algorithm for correcting the extant ozone historical record (described a paper currently in review) – particularly results measured at large slant column ozone amounts (e.g.: particularly at high latitudes in spring and fall). The inclusion of multiple results would obscure the basic intent of the paper and possibly create confusion in the community. Calculating the ozone absorption coefficients with new cross-sections is beyond the scope of this article and subject of another study which should be supported by WMO. To the best of our knowledge no new data have been submitted to the WOUDC using new cross-sections nor have the historical data been so corrected.

About the Dobson AD-CD difference: This work shows that higher levels of stray light lead to larger differences between AD and CD measurements. This fact is the same for all cross-sections. Redondas et al. (2014) have shown that for the ideal slit functions without stray light, the difference is somewhat lower for the Serdyuchenko (2014) cross-sections as compared with the results using other cross-sections.

**Specific comments**

**Page 1**

Line 20 – This needs to be done using Sedyuchenko cross-sections to be relevant and comparable to modern work.

Please see the response to the general comments.

Line 22 – I dispute that you have "evaluated" the error. The discrepancy between Dobson and Brewers as a result of their different assumptions is calculated but nothing here says what the deviation from the true value is.

Revised – "error" is replaced by "discrepancy". However, any difference between the actual height of the ozone layer and the assumption leads to an error in the measurements. The ozone climatology studies suggest that the ozone layer height is about 26 km and constant from -20 to 20 and then slopes toward the poles with the height around 17 km.

The authors believe that this assumption better matches with reality and is a good choice to be considered by both communities (Dobson and Brewer) as the height of the ozone layer. This leads to a reduction of discrepancies between Brewer and Dobson measurements and a more accurate ozone amount.

Line 24 Between 2008 and 2012 – this is quite misleading because you actually only use two distinct periods in 2008 and 2012 (Unless the description on page 11 is wrong?)

Description on page 11 is revised. All direct sun measurements available between 2008 and 2014 from three Dobsons and one double Brewer are used in this study.

Line 25 I can't see that you have shown this at all. You have shown the difference between the Dobson values and the double-Brewer values, but how have you actually attributed this difference to stray light? This is a serious defect that needs to be addressed.

The models developed in this study show that the documented instrumental stray light effect in Brewers causes a non-linearity in the modeled ozone measurements at large ozone slant paths consistent with the observations. In this study, the data from three Dobsons are compared to the measurements of a double-Brewer with a very low level of stray light. The discrepancies in the measurements clearly depend on ozone slant path which is an indicator of a stray light effect.

Line 30 I wouldn't say a "similar network" was introduced because of the more limited geographical coverage of Brewers even to today.

Revised – "similar" is removed.

**Page 2**

Line 15 Refer to Staehlin et al. GAW report

The reference is added to the manuscript.

Staehelin, J., J. B. Kerr, R. Evans, and K. Vanicek, (2003), COMPARISON OF TOTAL OZONE MEASUREMENTS OF DOBSON AND BREWER SPECTROPHOTOMETERS AND RECOMMENDED TRANSFER FUNCTIONS, *WMO/GAW Report No. 149*, World Meteorological Organization, Geneva, Switzerland.

Line 16 I think you need to be specific here – what fraction of the difference can be accounted for?

The following sentence has been added:

"They found a 3% drift over about a 10-year period (1988-1997) between the Arosa Dobson and Brewer total ozone series that remains unexplained."

Line 21 "properly" is not the right word, a lot of work has been done, eg at the RBCCE

Revised. Following statement is removed from the manuscript:

"the effect of the stray light on measurements at large solar zenith angles (SZA), have not been analyzed properly yet."

This paper reports the results of a physical model which demonstrates that the functional behavior of the non-linearity in ozone amounts at large slant ozone columns can be explained as being due to the measured stray light properties of the Brewer instruments and, by inference, a similar statement is made about the Dobson.

**Page 3**

Line 4 "large SZA and large TOC" – this is only true in the Northern Hemisphere. It is not true at all in Antarctica, which you use for your comparison. Was South Pole even a good choice for your study?

The South Pole is one of the most important sites to detect ozone recovery, trend analysis and satellite verifications. With ozone recovery in future more data measured at large ozone slant paths would be available which could cause an error in future studies. In addition, if the Dobson at that site is replaced with an instrument with a lower level of stray light, a false positive signal could be detected. There are six years of data collected by three Dobsons and one double Brewer which can be used for comparison. Using these data this study shows the dependence of the Dobson and Brewer difference to the ozone slant path which is an indicator of the impact of stray light.

**Page 5**

Line 24 It's fine to do the calculations using Bass-Paur so you can compare them to older work but you also have to do them using Serdyuchenko to be relevant to modern work, eg Redondas et al. 2014, Köhler et al. 2018)

The goal of this study is to show the effect of stray light at large ozone slant paths and also on effective ozone absorption coefficients for both Brewer and Dobson instruments. In our analysis the cross-section can be left as a variable and the whole process would be the same. Therefore, the authors believe that adding more numbers to the article would not be helpful and is beyond the scope of this study.

Line 29 the "relevant temperature" – you need to be explicit here – are you using the same temperature for both Dobsons and Brewers? Which is it? Otherwise won't this introduce a difference separate to what you're looking at?

The same temperature, -46.3, is used for both instruments. The manuscript is revised to clearly express this.

Line 25-27 To be clear, you are not going to use this approximation? (equation 18). You should be explicit.

This approximation ($\bar{\alpha}^{apx}$) is used for the values in table 3 rows 1 and 4 and also in table 4 columns 4, 6, and 8.

**Page 7**

Line 4 Why do you use theta_0 not just theta?

Revised. Theta_0 is replaced with theta.

Line 9 You say "it is important" but don't give any evidence as to why it's important. Evidently the Brewer algorithm doesn't think it's important

The fixed ozone layer height of 22 km, as used by Brewer retrieval, causes up to a 2.2 % difference at an air mass of 5.4 with the value retrieved with ozone layer height of 17 km (Figure 6). This shows that this parameter is important for large-solar-zenith-angle measurements, especially at higher latitude sites where the ozone layer height is close to 17 km based on ozone climatology studies.

Line 9 You say the "correct" value of the height of the ozone layer but don't show that the Dobson parameterisation is correct. I think you just mean that the Brewer and Dobson methods are different to each other and this will cause a slight difference in derived total ozone.

"correct" is replaced with "adjust" to prevent any confusion. No value is suggested by the authors as the correct value. But we believe that adopting a variable ozone layer height with latitude is in more agreement with ozone climatology. We suggest a constant value for -20 to 20 with slope toward the poles with 17 km at the poles.

Line 19 How do you know the stray light in a Dobson is similar to Mk IV and Mk II Brewers? Are you taking this from previous studies? This is one of my major confusions. I don't think you measured it?

No, we have not measured it. That statement is removed from the manuscript. However, the stray light levels suggested by Basher (1982) for several Dobson instruments based on the non-linearity in their measurements reveal that the level of stray light in Dobson instruments are comparable with measured stray light of Brewers MKII and MKIV.

Line24 You should mention that He-Cd laser has been used before and give the references (see Pulli et al. 2018)

References are added.

Pulli, T., T. Karppinen, S. Nevas, P. Kärhä, K. Lakkala, J. M. Karhu, M. Sildoja, A. Vaskuri, M. Shpak, F. Manoocheri, L. Doppler, S. Gross, J. Mes, and E. Ikonen, (2018), Out-of-Range Stray Light Characterization of Single-Monochromator Brewer Spectrophotometers, *Atmosphere-Ocean*, *56*(1), 1–11, doi:10.1080/07055900.2017.1419335.

Karppinen, T., A. Redondas, R. D. García, K. Lakkala, C. T. McElroy, and E. Kyrö, (2014), Compensating for the Effects of Stray Light in Single-Monochromator Brewer Spectrophotometer Ozone Retrieval, *Atmosphere-Ocean*, 1–8, doi:10.1080/07055900.2013.871499.

Kiedron, P., P. Disterhoft, and K. Lantz, (2008), NOAA-EPA Brewer network Stray Light Correction, NOAA Earth System Research Laboratory.

Line 24-25 You should give at least a very brief description of the experimental set-up.

For example, you should explain how you derive a slit-function from a single wavelength laser? What is the sensitivity of your detector? (This is important since you are measuring over such a wide dynamic range).

Revised. Following statement is added to the manuscript:

"The Brewer Mark III and IV can measure the wavelength range of 286.5 to 363 nm with 0.5 nm resolution. The Brewer slit function is characterized using a narrow band line source such as a laser as input source and scanning through all wavelengths. Measurements at 350 nm (not reported) have shown the slit function to be similar at all wavelengths in the Brewer measurement range. The slit function is reversed in wavelength space to account for the reciprocal nature of scanning the instrument v. scanning the wavelength of the line source."

**Page 8**

Line 15 You need to refer to Köhler et al. 2018.

The reference is added.

Lines 29-32 I am very confused here about what is what. In Figure 3 the slit functions are curved, not trapezoids. Where did this shape come from?

Caption of the figure was not accurate. Those are Dobson C-pair slit functions parameterized from Figure 1 of Komhyr (1993) and brought here as an example but are not being used for the analysis. The accurate description of the slit functions used for this study can be found in Table 1.

The caption has been corrected.

**Page 9**

Line 3 It seems you are not using the approximation in equation 18. Did you use a radiative transfer model?

In section 3.1 the approximation in equation 18 is used and the results are compared with the results from equation 17.

For section 3.2 the values calculated from equation 17 are used.

No, a radiative transfer model is not used in this study. In section 3.2 the solar spectrum at the surface, assuming constant amount of ozone in the atmosphere, is calculated using equation 1. Then, the Brewer and Dobson algorithm plus ideal and model slit functions are employed to retrieve the total ozone.

**Page 10**

Lines 8-9 I find this statement completely baffling. What do you mean by "measured slit functions"? What value of stray light are you suggesting WMO use?

Revised. Following statement is used instead of the previous one:

"It is advisable that the WMO assign a group to measure the Dobson stray light level at least for the reference instrument. Then, the absorption coefficients should be recalculated and recommended to be used instead of the values currently in use."

Line 15 I would like to see a plot showing what the Langely looked like without and without the stray light being added to the model

Following figure shows the Langley plot for Single Brewer with and without stray light. The plot is added to the manuscript:

"To calculate the ETC, the instrument absorption function using the solar spectrum (Chance and Kurucz 2010), Eqs. (1), (2) and retrieval algorithm of the Brewer (or Dobson) for an assumed constant amount ozone (325 DU in this study) is calculated and plotted as a function of ozone slant path. The best fit to the data with airmass less than 2 (less than 3 for the Dobson instruments) is found and extrapolated to zero airmass. Figure 5 shows the best fit to a single Brewer data:

[Figure]

**Figure 5: Example of Langley plot fitted to a modeled single Brewer data**

Line 3 It seems to me your results would imply the AD-CD correction should use an expression linear in mu rather than an average across the mu range?

Our model suggests that the AD-CD difference depends on the internal stray light level of the instrument and changes non-linearly above 800 DU OSP.

Line 15 You can't say "error" because you don't make any attempt to look at the what the true value is (for example by using South Pole ozonesonde data). You could call it a "discrepancy" between the Dobson and Brewer.

Revised. "discrepancy" is used instead of "error".

Line 30 Do you mean "February 2008 and December 2014" or is it actually meant to be "February 2008 to December 2014" ?

Revised. "February 2008 to December 2014" is correct.

Line 31 I don't think you can say "corrected" because you don't know that the Dobson value is any more correct than the Brewer value.

Revised. "adjusted" is used instead of "corrected".

**Page 12**

Line 9-14 It looks like a linear relation. Couldn't you then calculate the gradient and compare the value with your model? This is what I was expecting you to do to better finish off the study.

The comment is not clear. The stray light effect causes non-linearity in the measurements at large OSPs. There isn't a linear relation.

Line 24-26 It seems a bit curved. You need to calculate how close the measured values are to what you expect from your model.

The manuscript is revised and the following plots and statement are added:

"As it is shown in Fig. 9, the physical model developed in this study suggests $10^{-3.7}$, $10^{-4.1}$, and $10^{-4.0}$ level of stray light for Dobson #82, 42, and 80 respectively.

[Figure]

[Figure]

[Figure]

**Figure 9: The ratio of quasi-simultaneous observations of Dobson #82, 42 and 80 AD wavelengths to double Brewer #085 data at the South Pole as well as the ratio of the values retrieved from the physical model developed in this study with certain amounts of stray light to the true value assumed in the atmosphere suggesting the level of stray light in each individual Dobson instrument. The ETC values and ozone absorption coefficients are calculated for each model separately using Langley method. Note that the average difference between the Brewer and Dobson data with OSPs less than 800 DU has been used to scale the Dobson data first. Then the model with stray light level that better matches with the Dobson data has been found. The scaling factors used for Dobson #82, #42 and #80 are 1, 0.46 and 1.6 respectively."**

Line 28 "Physical model" but four lines later you say "mathematical model".
Revised. "Physical Model" is correct. (In reality it is both. The physics of the measurement is expressed in mathematical terms so that simulated observations can be produced.)

Line 30 Again, I don't think you have found anything about the "error" – only the difference between the Dobson and Brewer.

Revised. "difference" is used instead of "error".

Line 32 But does this 25% relate to a realistic value of Dobson stray light?

This is the error at 2000 DU OSP. Experimental data discussed by Basher (1982), Varotsos (1998) and Christodoulakis et al. (2015) suggest an even larger error at the same OSP indicating a higher level of stay light in the Dobsons being used for the measurements.

**Page 13**

Line 2 "like polar stations" would be better worded as "such as polar stations"

Revised.

Line 2 This is misleading because the study has considered South Pole data where ozone is very low in spring

The location doesn't really matter in the case of stray light. Wherever you have data with OSP (ozone slant path) larger than 800 DU you can see the effect of stray light. Even at South Pole there is enough data to show this effect.

Line 6 You say "stray light also can affect" but isn't this just a different way of expressing the same thing? (ie stray light will affect total ozone, which alternatively you could express as the effect on the absorption coefficients).

There are two effects we discussed in this article. First, the models show that the stray light causes non-linearity in the measurements at large ozone slant paths. This feature happens even if the ozone absorption coefficients are accurately calculated.

It is also shown that the stray light affects the calculation of the effective ozone absorption coefficients which leads to an error in the measurements even at low airmass. This error is reduced during calibration of the instruments by comparing the coincidence measurements with the reference instrument.

Following sentences are added to the manuscript:

"Recently, the measured slit functions and calculated coefficients were verified by measuring the slit functions of three Dobsons (#74, #64, and #83) at the Physikalisch-Technische Bundesanstalt (PTB) in Braunschweig in 2015 and at the Czech Metrology Institute (CMI) in Prague in 2016 within the EMRP ENV 059 project "Traceability for atmospheric total column ozone" (Köhler et al., 2018). Köhler et al. showed that the optical properties of these three Dobsons deviate from the specification described by G.M.B. Dobson. However, the AD pair ozone absorption coefficents derived from the new slit functions lead to less than a 1% deviation in total ozone column values."

Line 34 This calculation needs to be with Serdyuchenko cross-sections.
With ideal slit functions it was shown that the AD-CD difference minimizes with the Serdyuchenko cross-sections (Redondas, 2014) as compared to the results with other cross-sections. We show in this article that no matter which cross-sections are used, the AD-CD difference increases as the contribution of instrumental stray light level increases.

**Page 14**
Line 6 "high" should be "higher"
Revised.

Line 7 "low" should be "lower"
Revised.

Page 20 Table 3 Are the values for the single or double Brewer?
Both instruments are discussed in the table 3. "Single" for single Brewer and "Double" for double Brewer.

Page 23 Figure 2 Would this be better on a log scale as in Figure 1?
Revised.

Page 24 Figure 3 I am very confused about this figure. The shapes are cruved not straight lines. Did you measure these in the lab with the laser?

The caption of the figure is revised as discussed in previous comment (Page 8, line 29-32).

Page 29 Figure 8 How did you identify the outliers? (If they represent bad data, perhaps you shouldn't plot them?)

In each bin the values with differences larger than three standard deviations from the mean of the bin have been removed from the plot. This statement has been added to the caption of the figure.

REFERENCES

Köhler, U., Nevas, S., McConville, G., Evans, R., Smid, M., Stanek, M., Redondas, A., and Schönenborn, F.: Optical characterisation of three reference Dobsons in the ATMOZ Project – verification of G. M. B. Dobson's original specifications, Atmos. Meas. Tech., 11, 1989-1999, https://doi.org/10.5194/amt-11-1989-2018, 2018.

Koukouli, M. E., Zara, M., Lerot, C., Fragkos, K., Balis, D., van Roozendael, M., Allart, M. A. F., and van der A, R. J.: The impact of the ozone effective temperature on satellite validation using the Dobson spectrophotometer network, Atmos. Meas. Tech., 9, 2055-2065, https://doi.org/10.5194/amt-9-2055-2016, 2016.

T. Pulli, T. Karppinen, S. Nevas, P. Kärhä, K. Lakkala, J. M. Karhu, M. Sildoja, A. C6 Vaskuri, M. Shpak, F. Manoocheri, L. Doppler, S. Gross, J. Mes & E. Ikonen (2018) Out-of-Range Stray Light Characterization of Single-Monochromator Brewer Spectrophotometers, Atmosphere-Ocean, 56:1,1-11, DOI: 10.1080/07055900.2017.1419335

Staehelin, J., Kerr, J., Evans, K., and Vanicek, R.: Comparison Of Total Ozone Measurements of Dobson and Brewer spectrophotometers and Recommended Transfer Functions, WMO TD No. 1147, World Meteorological Organization, Global AtmosphereWatch, No. 149, available at: http://www.wmo.ch/web/arep/reports/gaw149.pdf., 2003.

Stübi, R., H. Schill, J. Klausen, L. Vuilleumier, and D. C. Ruffieux (2017), Reproducibility of total ozone column monitoring by the Arosa Brewer spectrophotometer triad, J Geophys. Res. Atmos., 122, 4735–4745, doi:10.1002/2016JD025735

Karppinen, T., A. Redondas, R. D. García, K. Lakkala, C. T. McElroy, and E. Kyrö, (2014), Compensating for the Effects of Stray Light in Single-Monochromator Brewer Spectrophotometer Ozone Retrieval, *Atmos. Ocean*, 1–8, doi:10.1080/07055900.2013.871499.

Christodoulakis, J., C. Varotsos, a. P. Cracknell, C. Tzanis, and A. Neofytos, (2015), An assessment of the stray-light in 25 years Dobson total ozone data at Athens, Greece, *Atmos. Meas. Tech.*, *8*, 3037–3046, doi:10.5194/amt-8-3037-2015.

Varotsos, C. A., (1998), Technical note On the influence of stray light on total ozone measurements made with Dobson spectrophotometer no. 118 in Athens, Greece, *Int. J. Remote Sens.*, *19*(17), 3307–3315, doi:10.1080/014311698213993.

Kiedron, P., P. Disterhoft, and K. Lantz, (2008), NOAA-EPA Brewer network Stray Light Correction, NOAA Earth System Research Laboratory.

---

## Author Response (AR2)

**Omid Moeini et al. (omidmns@yorku.ca)**

The authors would like to thank the editor and reviewer for the useful review which we have gratefully considered in improving the paper.

The analyses are repeated using Serdyuchenko et al. (2014) cross-sections and the results are reported by

10 updating the corresponding tables and figures.

[revised manuscript text omitted]